# Distinct fibroblast subsets regulate lacteal integrity through YAP/TAZ-induced VEGF-C in intestinal villi

Seon Pyo Hong[1,8], Myung Jin Yang[2,8], Hyunsoo Cho [1,3,8], Intae Park[1], Hosung Bae [1], Kibaek Choe[4], Sang Heon Suh[1,3], Ralf H. Adams [5], Kari Alitalo[6], Daesik Lim [7] & Gou Young Koh [1,2,3 ✉]

Emerging evidence suggests that intestinal stromal cells (IntSCs) play essential roles in maintaining intestinal homeostasis. However, the extent of heterogeneity within the villi stromal compartment and how IntSCs regulate the structure and function of specialized intestinal lymphatic capillary called lacteal remain elusive. Here we show that selective hyperactivation or depletion of YAP/TAZ in PDGFRβ⁺ IntSCs leads to lacteal sprouting or regression with junctional disintegration and impaired dietary fat uptake. Indeed, mechanical or osmotic stress regulates IntSC secretion of VEGF-C mediated by YAP/TAZ. Single-cell RNA sequencing delineated novel subtypes of villi fibroblasts that upregulate *Vegfc* upon YAP/TAZ activation. These populations of fibroblasts were distributed in proximity to lacteal, suggesting that they constitute a peri-lacteal microenvironment. Our findings demonstrate the heterogeneity of IntSCs and reveal that distinct subsets of villi fibroblasts regulate lacteal integrity through YAP/TAZ-induced VEGF-C secretion, providing new insights into the dynamic regulatory mechanisms behind lymphangiogenesis and lymphatic remodeling.

[1] Center for Vascular Research, Institute for Basic Science (IBS), Daejeon 34141, Republic of Korea. [2] Biomedical Science and Engineering Interdisciplinary Program, Korea Advanced Institute of Science and Technology (KAIST), Daejeon 34141, Republic of Korea. [3] Graduate School of Medical Science and Engineering, KAIST, Daejeon 34141, Republic of Korea. [4] Graduate School of Nanoscience and Technology, KAIST, Daejeon 34141, Republic of Korea. [5] Department of Tissue Morphogenesis, Max Planck Institute for Molecular Biomedicine, and University of Münster, D-48149 Münster, Germany. [6] Translational Cancer Biology Program and Wihuri Research Institute, Biomedicum Helsinki, University of Helsinki, Helsinki 00290, Finland. [7] Department of Biological Sciences, KAIST, Daejeon 34141, Republic of Korea. [8] These authors contributed equally: Seon Pyo Hong, Myung Jin Yang, Hyunsoo Cho. ✉email: gykoh@kaist.ac.kr

Each small intestinal villus contains a highly specialized lymphatic capillary called a lacteal, which is essential for dietary fat uptake and gut immune surveillance[1–3]. The lymphatic endothelial cells (LECs) that make up lacteals constantly but slowly proliferate under vascular endothelial growth factor (VEGF)-C–VEGF receptor (VEGFR)3 and delta-like 4 (DLL4)-Notch signaling, whereas most LECs forming the lymphatic vessels (LVs) in other organs are relatively quiescent in adults[4,5]. Moreover, adrenomedullin-calcitonin receptor and VEGF-A-VEGFR2 signaling have been demonstrated as regulators of lacteal integrity and function, and vascular endothelial (VE)-cadherin is also required for lacteal morphogenesis and maintenance[6–8]. Renewal capacity is unique to certain organ-specific LECs, which implies that distinct regulatory processes may support lacteals to cope with the dynamic microenvironment and highly repetitive mechanical forces of intestinal peristalsis[1–3]. A lacteal is surrounded by the villus stroma, which is composed of capillaries, pericytes, smooth muscle cells (SMCs), fibroblasts, immune cells, and extracellular matrix residing under the rapidly renewing intestinal epithelial cells[1–3]. In this milieu, a subset of SMCs has been suggested as an important source of VEGF-C for lacteal maintenance[5]. On the other hand, emerging evidence suggests that the mesenchymal compartment of the intestinal stroma provides Wnt ligand for intestinal stem cells and is important in maintaining gut homeostasis by tightly coordinating the renewal capacity of intestinal stem cells[9–12]. Indeed, mesenchymal stromal cells are also being recognized to have diverse roles in microenvironmental remodeling in various organs[13–16]. However, although lacteals are compactly surrounded by mesenchymal stromal cells, their contributions to lacteal maintenance remain unknown.

The mammalian Hippo signaling pathway is the key regulator that controls various biological processes including cellular differentiation, proliferation, tissue homeostasis, and stem cell renewal[17–19]. The key components of the Hippo kinase cascade are LATS1 and LATS2 (LATS1/2), and the final transcriptional regulators YAP and TAZ (YAP/TAZ) primarily bind to TEA domain transcription factors (TEAD) to exert their activities[18]. Recent studies have demonstrated that the Hippo pathway regulates stromal cells, including blood endothelial cells (BECs), LECs, SMCs, and pericytes, to promote tissue development and function in a highly context-dependent manner[20–24]. Indeed, intimate Hippo-dependent crosstalk among these cell types is essential for tissue-specific formation and specialization of vessels, as well as their remodeling[20–24]. YAP/TAZ are regulated by mechanical stress, and they are well-known mediators and effectors of mechanical cues for the regulation of cell proliferation and differentiation[25–29]. In particular, mesenchymal stromal cells in intestinal villi are placed under high levels of repetitive mechanical stress and extreme variations in osmotic pressure[1–3] that can modulate YAP/TAZ.

Here, we explored the role of intestinal villi stromal cells by generating PDGFRβ-specific YAP/TAZ hyperactivation or depletion mouse models. We show that PDGFRβ+ intestinal stromal cells (IntSCs) play pivotal roles in lacteal maintenance and function by YAP/TAZ-dependent secretion of VEGF-C. We demonstrate that YAP/TAZ of PDGFRβ+ IntSCs integrate mechanical cues and VEGF-C secretion to support lacteal integrity. Using single-cell RNA sequencing, we reveal the heterogeneity of PDGFRβ+ IntSCs by identifying five distinct clusters of fibroblasts in addition to enteric SMCs and mural cells. Furthermore, we visualize the unique location of each subtype using differentially expressed marker genes identified in our analysis. Our results show that certain subsets of fibroblasts surrounding a lacteal secrete VEGF-C upon YAP/TAZ activation. These findings propose the novel and dynamic regulatory

mechanism of lacteal maintenance and function by distinct subsets of fibroblasts, which constitute the peri-lacteal microenvironment.

## Results

### YAP/TAZ hyperactivation in PDGFRβ+ IntSCs promotes aberrant lacteal spouting and branching network formation in the small intestinal villi.

LATS1/2 constitute the core of Hippo pathway, which inhibits YAP/TAZ activities through phosphorylation-dependent cytoplasmic retention and degradation[18]. To gain insights into the role of Hippo pathway in PDGFRβ+ cells, we hyperactivated YAP/TAZ by deleting both Lats1/2 in these cells in a tamoxifen-dependent manner using the Lats1/2^iΔPβC mutants, which were generated by crossing Pdgfrb-Cre-ER^T2 mouse[30] with Lats1^fl/fl/Lats2^fl/fl mouse[31,32] (Fig. 1a). Cre-ER^T2-negative but flox/flox-positive littermates were defined as wild-type (WT) mice in each experiment.

The most striking finding was observed along the whole small intestine including duodenum, jejunum, and ileum of Lats1/2^iΔPβC mice, which revealed aberrant lacteal sprouting and branching network formation, while lacteals of WT mice showed single, blunt-ended, and finger-like round tube that is representative of a normal lacteal (Fig. 1b, c). The surface area and length of lacteals increased by ~1.6 and ~1.3-fold, respectively, while villi width was reduced by ~23% in all portions of small intestine in Lats1/2^iΔPβC mice compared with WT; villi length was not significantly changed (Fig. 1b, c). Detailed analyses of lacteals in jejunum revealed presence of 8–9 lymphatic sprouts and 11–12 lymphatic branching networks in 100 μm length of each lacteal of Lats1/2^iΔPβC mice, which were rarely visible in WT mice (Fig. 1d, e). Besides, numbers of Prox1+ LECs and EdU+Prox1+ proliferating LECs increased by 2.4-fold and 18.1-fold in the lacteals of Lats1/2^iΔPβC mice compared with those of WT mice (Fig. 1f, g), indicating that the lacteal LECs are actively proliferating in Lats1/2^iΔPβC mice.

When we examined the expressions and distributions of YAP/TAZ in the mouse intestinal villi, YAP/TAZ proteins were moderately enriched in PDGFRβ+ IntSCs as a nuclear (N) and cytoplasmic (C) localization type (N = C type) in WT (Supplementary Fig. 1a, b). In comparison, their levels were increased and distribution pattern was changed to a predominant nuclear localization type (N > C type) in PDGFRβ+ IntSCs of Lats1/2^iΔPβC mice (Supplementary Fig. 1a, b). Moreover, to confirm that the LATS1/2 deletion induces aberrant lacteal phenotypes via YAP/TAZ, we observed the lacteal phenotypes using Lats1/2-Yap/Taz^iΔPβC mice (Supplementary Fig. 2a). The aberrant sprouting and branching phenotypes in lacteals of Lats1/2^iΔPβC mice were restored in Lats1/2-Yap/Taz^iΔPβC mice (Supplementary Fig. 2b). This result indicates that LATS1/2 deletion alters lacteal morphogenesis through YAP/TAZ.

### YAP/TAZ hyperactivation in PDGFRβ+ IntSCs alters lacteal LEC junctions and impairs lipid drainage into lacteals.

LVs are lined by LECs and they consist of initial lymphatic capillaries with button-like junctions and collecting LVs with zipper-like junctions[33]. Similar to button-like junctional organization of LECs in lymphatic capillaries of peripheral organs[33], lacteal junctions consisted mostly of discontinuous button-like (~54%) junctions (Fig. 2a, b). However, the proportion of button-like junction was reduced by ~83%, while the proportion of zipper-like junction increased by ~4.0-fold in lacteals of Lats1/2^iΔPβC mice (Fig. 2a, b). Further analysis with transmission electron microscopy revealed aberrantly overlapped, large vacuole-containing complicated and thickened lacteal LECs with closed junctions, and chylomicrons were rarely detected in lacteal lumen of Lats1/2^iΔPβC mice. On the

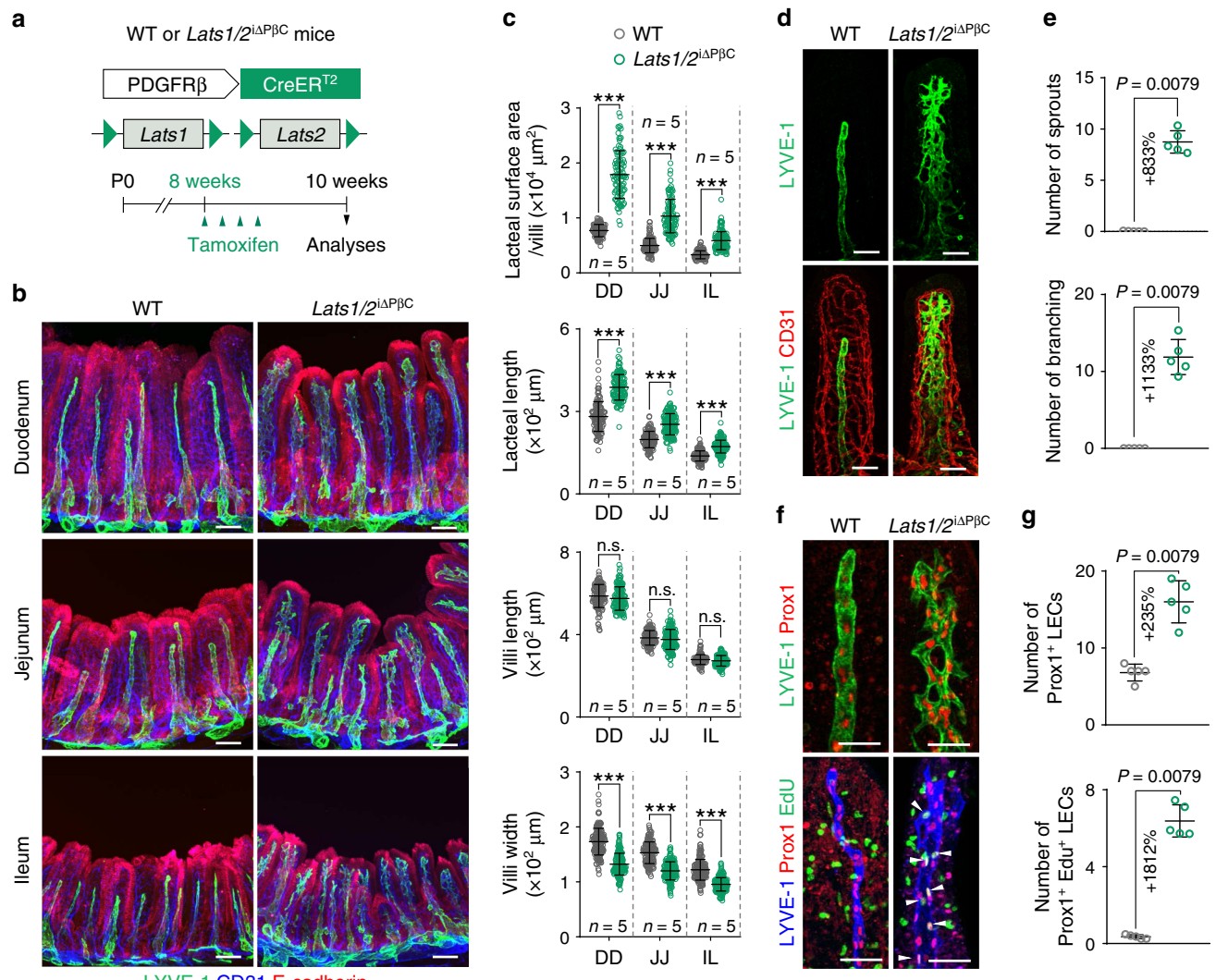

**Fig. 1 YAP/TAZ hyperactivation in PDGFRβ⁺ IntSCs induces lacteal sprouting and branching. a** Diagram depicting the generation of *Lats1/2*[iΔPβC] mouse and PDGFRβ⁺ cell-specific depletion of *Lats1/2* in 8-week-old mice and analyses at 2 weeks later. **b, c** Representative images of LYVE-1⁺ lacteals and CD31⁺ capillary plexus under the E-cadherin⁺ epithelial cells in intestinal villi of the indicated part of small intestine and comparisons of the lacteal surface area per villi, lacteal length, villi length, and villi width in the duodenum (DD), jejunum (JJ), and ileum (IL) in WT and *Lats1/2*[iΔPβC] mice. Dots indicate values of 101–217 villi/group in $n = 5$ mice/group pooled from three independent experiments. Horizontal bars indicate mean ± SD and *** $P < 0.0001$ versus WT by two-tailed Mann–Whitney $U$ test. n.s., not significant. Scale bars, 100 μm. **d, e** Representative images and comparisons of the number of lacteal sprouts and branches per 100 μm of lacteal length in WT and *Lats1/2*[iΔPβC] mice. Each dot indicates a mean value of 10–20 villi/mouse and $n = 5$ mice/group pooled from three independent experiments. Horizontal bars indicate mean ± SD and $P$ value versus WT by two-tailed Mann–Whitney $U$ test. Scale bars, 50 μm. **f, g** Representative images and comparisons of the number of Prox1⁺ lymphatic endothelial cells (LECs) and Prox1⁺EdU⁺ proliferative LECs (white arrowheads) per 100 μm of lacteal length in WT and *Lats1/2*[iΔPβC] mice. Each dot indicates a mean value of 10–20 villi/mouse and $n = 5$ mice/group pooled from three independent experiments. Horizontal bars indicate mean ± SD and $P$ value versus WT by two-tailed Mann–Whitney $U$ test. Scale bars, 50 μm.

other hand, lacteal junctions were paracellularly open and chylomicrons were observed inside and outside of the lacteal lumen in WT mice (Fig. 2c). These results imply that the transport of chylomicrons into the lacteal is largely impaired in *Lats1/2*[iΔPβC] mice compared with WT.

To ensure this finding, we measured lipid clearance from the lamina propria of small intestinal villi into the lacteal after intraluminal loading of boron-dipyrromethene fluorescent-conjugated fatty acid (BODIPY-FA) using an intravital imaging system[34]. Normalized residual signals of BODIPY-FA in the lamina propria at 36 min and 46 min after BODIPY-FA loading were ~20–30% higher in *Lats1/2*[iΔPβC] mice compared with WT mice (Fig. 2d–f). Moreover, oil red O staining of *Lats1/2*[iΔPβC]

mice small intestine at 12 h after olive oil gavage revealed 5.5-fold increased accumulation of lipid in the lamina propria compared with WT mice (Fig. 2g–i). These results indicate that alterations in lacteal junction is accompanied by impaired dietary lipid uptake through the lacteal in *Lats1/2*[iΔPβC] mice.

To delineate the cause of lacteal impairment, we performed detailed morphological analysis of the intestinal villi. Lacteals are surrounded by longitudinal bundles of SMCs that facilitate transport of lymph containing chylomicron by periodic squeezing under the control of autonomic nervous system[1,3,34]. Notably, alignment of villus SMCs was skewed, distorted, and even aberrantly intermingled with the lacteals, and their coverage area was decreased by 55.9% in *Lats1/2*[iΔPβC] mice compared with WT

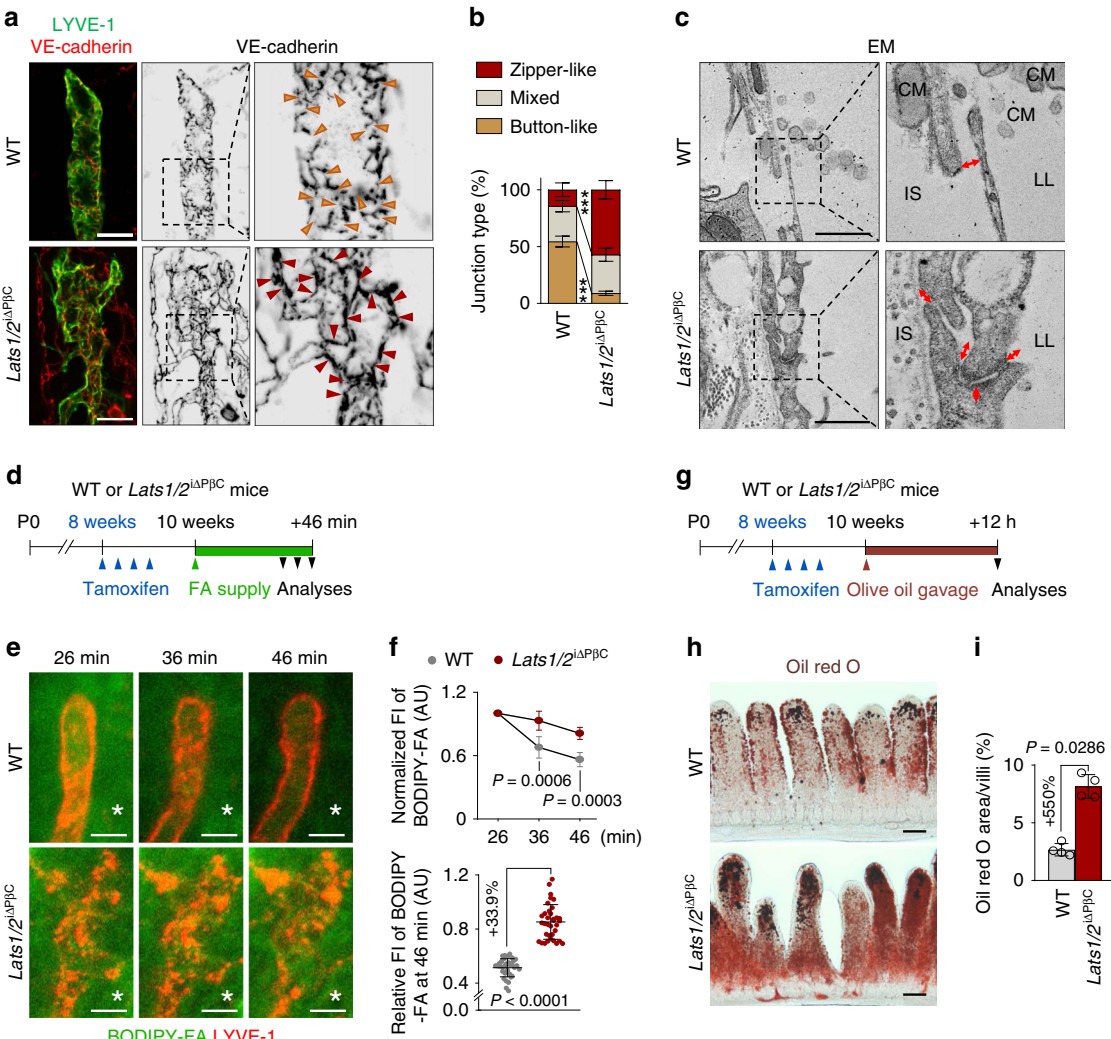

**Fig. 2 Button-to-zipper junctional transition and impaired lipid absorption in *Lats1/2*[iΔPβC] mice. a, b** Representative images and comparison of VE-cadherin[+] lymphatic endothelial cell (LEC) junctions of LYVE-1[+] lacteals in WT and *Lats1/2*[iΔPβC] mice. Black dotted box is magnified in the right panel and arrowheads indicate the lacteal junctional pattern that is dominant in each group with the same color described in **b**. Horizontal bars of each colored segment represent mean ± SD of 5–10 villi/mouse and $n = 5$ mice/group pooled from three independent experiments. *** $P < 0.0001$ versus WT by two-way ANOVA with Holm–Sidak's multiple comparisons test. Scale bars, 20 μm. **c** Representative transmission electron micrographs (EM) of a lacteal in WT and *Lats1/2*[iΔPβC] mice. Black dotted box is magnified in the right panel. LL, lacteal lumen; CM, chylomicron; IS, interstitium of lamina propria. Similar findings were observed in $n = 5$ mice/group from two independent experiments. Scale bars, 1 μm. **d** Diagram depicting the schedule of intravital imaging of lipid clearance from lamina propria via lacteals after supplying the BODIPY-fatty acid (FA) into the WT or *Lats1/2*[iΔPβC] mice and their analyses until after 46 min. **e, f** Representative intravital images and comparisons of the drainage of BODIPY-FA from the lamina propria (white asterisks) through the LYVE-1[+] lacteal in WT and *Lats1/2*[iΔPβC] mice. Each dot indicates a value of 4–8 villi/mouse and $n = 7$ (WT) or $n = 8$ (*Lats1/2*[iΔPβC]) mice pooled from four independent experiments. Horizontal bars indicate mean ± SD and $P$ value versus WT by two-tailed Mann–Whitney $U$ test. FI, fluorescence intensity; AU, arbitrary unit. Scale bars, 20 μm. **g** Diagram depicting the schedule of olive oil gavage at 12 h prior to the intestinal sampling in WT or *Lats1/2*[iΔPβC] mice. **h, i** Representative images and comparison of Oil red O staining in intestinal villi of WT and *Lats1/2*[iΔPβC] mice. Each dot indicates a mean value of 5–7 villi/mouse and $n = 4$ mice/group pooled from two independent experiments. Horizontal bars indicate mean ± SD and $P$ value versus WT by two-tailed Mann–Whitney $U$ test. Scale bars, 100 μm.

mice (Supplementary Fig. 3a–c). Also, while VEGFR3 expression in lacteal was increased by 1.6-fold, no significant change in VEGFR2[+] capillary plexus density was observed in intestinal villi of *Lats1/2*[iΔPβC] mice compared with WT mice (Supplementary Fig. 3d, e). Although the overall shape of the PGP9.5[+] neural network was slightly altered, the proportion of immune cells including CD3[+] T cells and F4/80[+] macrophages and VE-cadherin[+] alignment and junctional pattern of the BECs of capillary plexus in the lamina propria were not visibly altered in *Lats1/2*[iΔPβC] mice compared with WT mice (Supplementary Fig. 3d–g). In addition, F4/80[+] macrophages, previously

suggested as source of VEGF-C[35], showed no significant changes in their numbers between WT and *Lats1/2*[iΔPβC] mice. (Supplementary Fig. 4a–c). Furthermore, no notable alterations were detected in LVs of other organs including small intestinal submucosa, ear skin, tracheal mucosa, or central tendon of the diaphragm (Supplementary Fig. 5a–c). Collectively, these results suggest that lacteal phenotypes are primary effect by hyperactivation of YAP/TAZ in PDGFRβ[+] IntSCs.

**VEGF-C is upregulated in PDGFRβ[+] IntSCs of *Lats1/2*[iΔPβC] mice.** These findings led us to investigate how activation of

YAP/TAZ in PDGFRβ+ IntSCs causes lacteal spouting, branching, and proliferation. We generated $tdTomato^{rPβC}$ reporter mice by crossing $Pdgfrb$-Cre-ER$^{T2}$ mouse with $Rosa26$-$tdTomato$ reporter mouse to trace PDGFRβ+ cells in the small intestinal villi, where we observed confined and distinct distributions of PDGFRβ+ signals in subsets of longitudinal SMCs surrounding the lacteal, fibroblasts, and pericytes along the capillary network in the small intestinal stroma (Supplementary Fig. 6a–c). We then generated $Lats1/2^{iΔ}$-$tdTomato^{rPβC}$ mouse by crossing $tdTomato^{rPβC}$ mouse with the $Lats1^{fl/fl}/Lats2^{fl/fl}$ mouse and isolated PDGFRβ+ IntSCs from the small intestine of $tdTomato^{rPβC}$ and $Lats1/2^{iΔ}$-$tdTomato^{rPβC}$ mice for transcriptomic analysis with focus on comparison of secretory proteins (Supplementary Fig. 6d, e). Of note, heat map of RNA sequencing data demonstrated that genes including $Wisp2$ (145.1-fold), $Tnfsf15$ (14.2-fold), $Bdnf$ (8.9-fold), $Ebi3$ (8.4-fold), $Angpt2$ (6.3-fold), $Vegfc$ (6.1-fold) and $Ctgf$ (5.9-fold) were highly upregulated among the top 23 differentially expressed secretory molecules between $tdTomato^{rPβC}$ and $Lats1/2^{iΔ}$-$tdTomato^{rPβC}$ mice (Supplementary Fig. 6f). To further delineate the secretory molecule responsible for lacteal sprouting, we employed sprouting lymphangiogenesis assay using the spheroidal aggregates of LECs[36]. While VEGF-C alone induced robust sprouting of LEC spheroid, WISP2, TNFSF15, BDNF, EBI3, or ANGPT2 exhibited no significant effect on sprouting regardless of the presence or absence of VEGF-C (Supplementary Fig. 6g, h), implying that VEGF-C is likely the main factor responsible for the lacteal spouting. Indeed, mRNA levels of lymphangiogenic factors ($Vegfc$, 3.6-fold; and $Vegfd$, 2.4-fold) and VEGF-C activation factor ($Ccbe1$, 2.8-fold) were increased, although $Vegfa$, $Adamts3$, $Tgfb$, and $Ifng$ were not significantly altered in the small intestinal lysates of $Lats1/2^{iΔPβC}$ mice compared with WT mice (Supplementary Fig. 7a). Moreover, gene sets related to actin filament branching, apical surface cell polarity, and maintenance of cell polarity were remarkably altered according to gene set enrichment analysis of the isolated small intestinal LECs in $Lats1/2^{iΔ}$-$tdTomato^{rPβC}$ mice compared with those of $tdTomato^{rPβC}$ mice (Supplementary Fig. 7b). Ingenuity Pathway Analysis also revealed changes that were mostly related to the endothelial growth signaling and cell cytoskeleton regulation in small intestinal LECs of $Lats1/2^{iΔ}$-$tdTomato^{rPβC}$ mice (Supplementary Fig. 7c). Overall, our data indicates that VEGF-C level is not only upregulated but it could also be the main factor for lacteal phenotypes upon YAP/TAZ hyperactivation in the PDGFRβ+ IntSCs.

**VEGFR3 blockade completely abrogates aberrant lacteal sprouting and hyperbranching in $Lats1/2^{iΔPβC}$ mice.** As $Vegfc$ was highly upregulated in PDGFRβ+ intestinal stromal cells by YAP/TAZ hyperactivation, we sought to determine whether aberrant lacteal sprouting and hyperbranching are dependent on the major lymphangiogenic signaling, VEGF-C–VEGFR3 pathway. To interfere with this signal, we intraperitoneally (i.p.) injected $10^{12}$ viral particles of adeno-associated viral vectors (AAVs) encoding soluble VEGFR3 (AAV–mVEGFR3$_{1-4}$-Ig) or control vector (AAV–mVEGFR3$_{4-7}$-Ig)[37] on the day of tamoxifen delivery and examined the lacteal phenotypes of WT and $Lats1/2^{iΔPβC}$ mice 2 weeks later (Fig. 3a). The vector-encoded VEGFR3 protein was still detected in the serum at 2 weeks after AAV delivery (Fig. 3b). Strikingly, aberrant lacteal phenotypes of $Lats1/2^{iΔPβC}$ mice including increased Prox1+ lacteal LEC number, hypersprouting, and hyperbranching were completely blocked by administration of AAV–mVEGFR3$_{1-4}$-Ig (Fig. 3c–e). Moreover, lacteal LEC junctional alteration from predominant button-like pattern to zipper-like pattern in $Lats1/2^{iΔPβC}$ mice was completely blocked (Fig. 3f, g). In comparison, our approach

to block VEGFR2 signaling by α-VEGFR2 blocking antibody, DC101 (50 mg/kg every alternative day for 2 weeks), could only partially restore the aberrant lacteal sprouting and branching, but almost no changes were observed in the number of lacteal Prox1+ LECs and lacteal junctional transition compared with IgG control in $Lats1/2^{iΔPβC}$ mice (Supplementary Fig. 8a–f). These results indicate that activation of VEGF-C–VEGFR3 signaling plays a greater role than VEGFR2 signaling in aberrant lacteal sprouting and branching led by YAP/TAZ hyperactivation in PDGFRβ+ IntSCs.

**YAP/TAZ-induced VEGF-C secretion by PDGFRβ+ IntSCs could contribute to lacteal maintenance and lipid absorption.** To address the roles of YAP/TAZ in PDGFRβ+ IntSCs, we generated $Yap/Taz^{iΔPβC}$ mouse by crossing a $Pdgfrb$-Cre-ER$^{T2}$ mouse with a $Yap^{fl/fl}/Taz^{fl/fl}$ mouse[38] (Fig. 4a). YAP/TAZ depletion in PDGFRβ+ IntSCs was confirmed by immunohistochemical analysis (Supplementary Fig. 9a, b). Notably, lacteal length was decreased by ~35.6% while lacteal width was relatively increased by ~1.3-fold, but there were no other major alterations in the overall structure of the intestinal villi at 3 months after tamoxifen delivery in $Yap/Taz^{iΔPβC}$ mice compared with WT mice (Fig. 4b, c). Moreover, the lacteal junctional pattern of $Yap/Taz^{iΔPβC}$ mice exhibited ~83% decreased proportion of button-type, while the proportion of zipper-type was increased by ~4.0-fold compared with WT mice (Fig. 4d, e). However, the junctional pattern of villus blood capillary was not altered (Supplementary Fig. 9c), and no remarkable changes in LVs of other organs including small intestinal submucosa, ear skin, tracheal mucosa, and diaphragm were observed (Supplementary Fig. 10a–c). Nonetheless, plasma triglyceride concentration was reduced by 29.7–54.1% at 1~3 h after olive oil gavage in $Yap/Taz^{iΔPβC}$ mice compared with WT mice (Fig. 4f). Importantly, $Vegfc$ mRNA was decreased by 44.9%, while no changes in mRNA of $Vegfd$, $Vegfa$, $Ccbe1$, $Adamts3$, $Tgfb$, and $Ifng$ were observed in the small intestinal lysates of $Yap/Taz^{iΔPβC}$ mice compared with WT mice (Fig. 4g and Supplementary Fig. 9d). We also investigated whether lacteal dysfunction in $Yap/Taz^{iΔPβC}$ mice has an effect on dietary lipid absorption in $Yap/Taz^{iΔPβC}$ mice with high-fat diet (HFD) challenge for 8 weeks. While we observed no difference in weight gain during normal chow (NC) diet, $Yap/Taz^{iΔPβC}$ mice gained significantly less weight, exhibited less fat mass and less plasma triglycerides, and had less ectopic fat accumulation in the liver compared with WT mice (Supplementary Fig. 11a–g). These findings suggest that $Yap/Taz^{iΔPβC}$ mice have impaired lipid absorption, presumably due to decreased VEGF-C and lacteal dysfunction.

**YAP/TAZ-induced VEGF-C secretion from PDGFRβ+ IntSCs is regulated by mechanical cues and osmotic stress.** Intestinal villi are constantly exposed not only to high mechanical forces by peristaltic contractility but also to dynamically changing osmolarity during food digestion and absorption[1,3,4,34]. Of note, villi subepithelial fibroblasts have been suggested to respond to such high strain[1,3,4,34]. Because YAP/TAZ serve as mechanosensors and mechanotransducers[18,19,39], we analyzed whether YAP/TAZ are dictated by the mechanical stimuli for the secretion of VEGF-C in fibroblasts. We mechanically stretched (4% of linear stretch at 10 cycles/min) primary cultured mouse embryonic fibroblasts (MEFs) for 3 h to mimic the mechanical stimulus induced by intestinal peristalsis. This stimulus induced nuclear localizations of YAP/TAZ and increased the mRNA expressions of $Vegfc$ (~2.0-fold) as well as YAP/TAZ target genes such as $Ctgf$ (~5.7-fold) and $Ankrd1$ (~2.6-fold) (Supplementary Fig. 12a, b). In

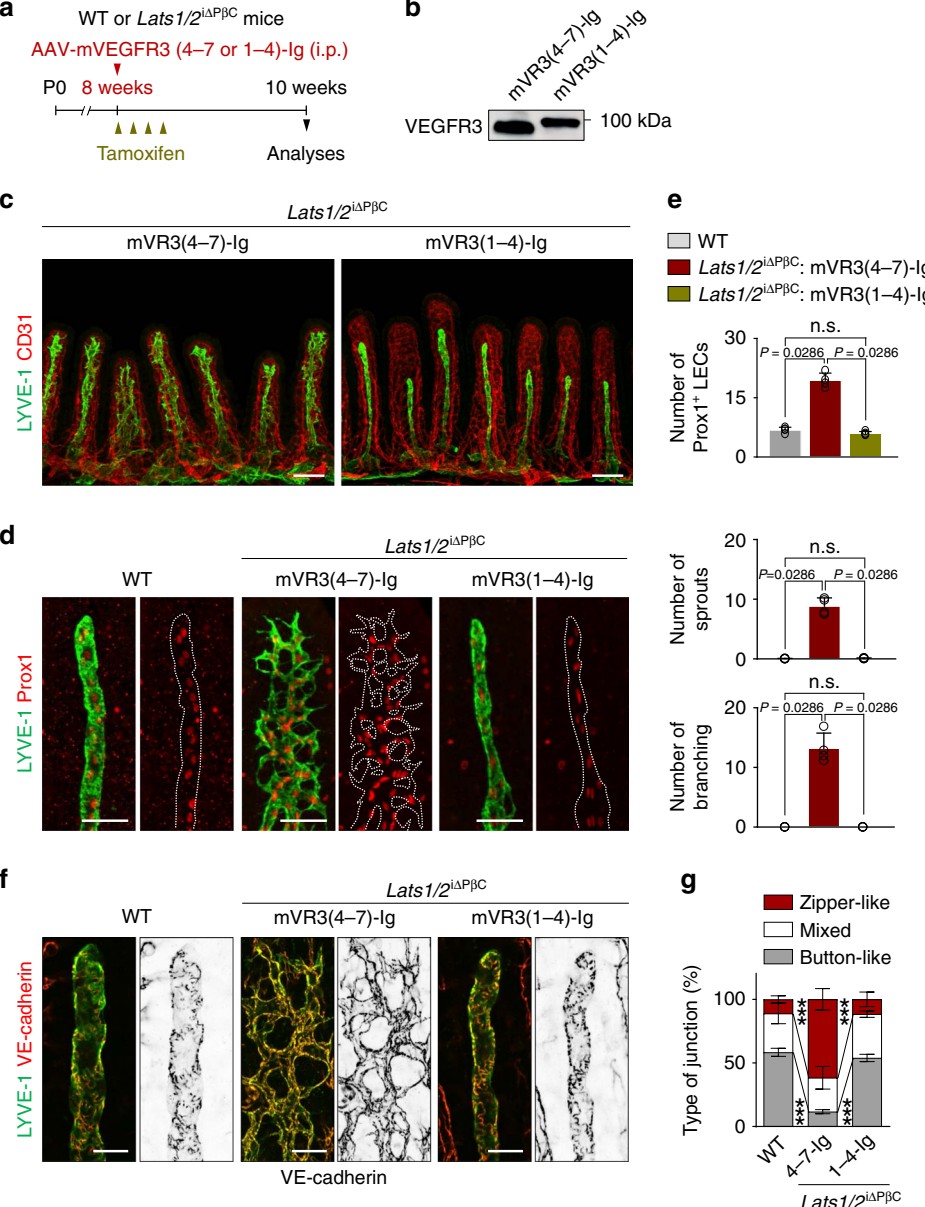

**Fig. 3 VEGFR3 blockade abrogates lacteal sprouting and branching in *Lats1/2*[iΔPβC] mice. a** Diagram for analyses of adult WT and *Lats1/2*[iΔPβC] mice two weeks after single intraperitoneal (i.p.) injection of AAV–mVEGFR3(4-7)-Ig (mVR3(4-7)-Ig, control) or AAV–mVEGFR3(1-4)-Ig (mVR3(1-4)-Ig, VEGFR3 blockade). **b** Representative immunoblot analysis showing mVEGFR3-Ig proteins in serum two weeks after single i.p. injection of AAV–mVEGFR3(4-7)-Ig or AAV–mVEGFR3(1-4)-Ig (1 × 10¹² viral particles in 150–200 µl PBS). kDa, kilodalton. Similar findings were observed in $n = 4$ mice/group from two independent experiments. **c–e** Representative images of CD31⁺ capillary plexus and LYVE-1⁺ lacteals in the villi of *Lats1/2*[iΔPβC] mice treated with mVR3(4-7)-Ig or mVR3(1-4)-Ig and comparisons of the number of Prox1⁺ lymphatic endothelial cells (LECs), lacteal sprouts, and lacteal branches of LYVE-1⁺ lacteals per 100 µm of lacteal length in the villi of WT and *Lats1/2*[iΔPβC] mice treated with mVR3(4-7)-Ig or mVR3(1-4)-Ig. White dotted-line outlines the lacteal. Each dot indicates a mean value of 10–20 villi/mouse and $n = 5$ mice/group pooled from two independent experiments. Horizontal bars indicate mean ± SD and *P* value versus WT or mVR3(4-7)-Ig by two-tailed Mann–Whitney *U* test. n.s., not significant. Scale bars, **c** 100 µm; **d** 50 µm. **f**, **g** Representative images and comparison of VE-cadherin⁺ LEC junctions of LYVE-1⁺ lacteals in the villi of WT and *Lats1/2*[iΔPβC] mice treated with mVR3 (4-7)-Ig or mVR3(1-4)-Ig. Horizontal bars of each colored segment represent mean ± SD of 5–10 villi/mouse and $n = 5$ mice/group pooled from two independent experiments. ***$P < 0.0001$ versus WT or mVR3(4-7)-Ig treated *Lats1/2*[iΔPβC] mice by two-way ANOVA with Holm–Sidak's multiple comparisons test. Scale bars, 25 µm.

contrast, high osmolarity (sorbitol 0.4 M) induced cytoplasmic localizations of YAP/TAZ[40] and attenuated the expresions of *Vegfc* (55.4%), *Ctgf* (80.9%), and *Ankrd1* (77.7%) in the MEFs (Supplementary Fig. 12c, d). These findings led us to corroborate the molecular interaction between YAP/TAZ and the *Vegfc* gene. We found several consensus TEAD-binding sequence (GGAAT) in the *Vegfc* promoter locus (Fig. 5a)[24]. To further validate these

regions as TEAD-binding sites, we performed chromatin immunoprecipitation (ChIP) analysis followed by q-PCR with primers encompassing indicated TEAD-binding sequences (Fig. 5a). Our ChIP q-PCR results revealed a significant endogenous binding of the TEAD protein near the TEAD-binding sequence (termed R3) found approximately 2.2 kb upstream from the *Vegfc* transcription start site (Fig. 5b).

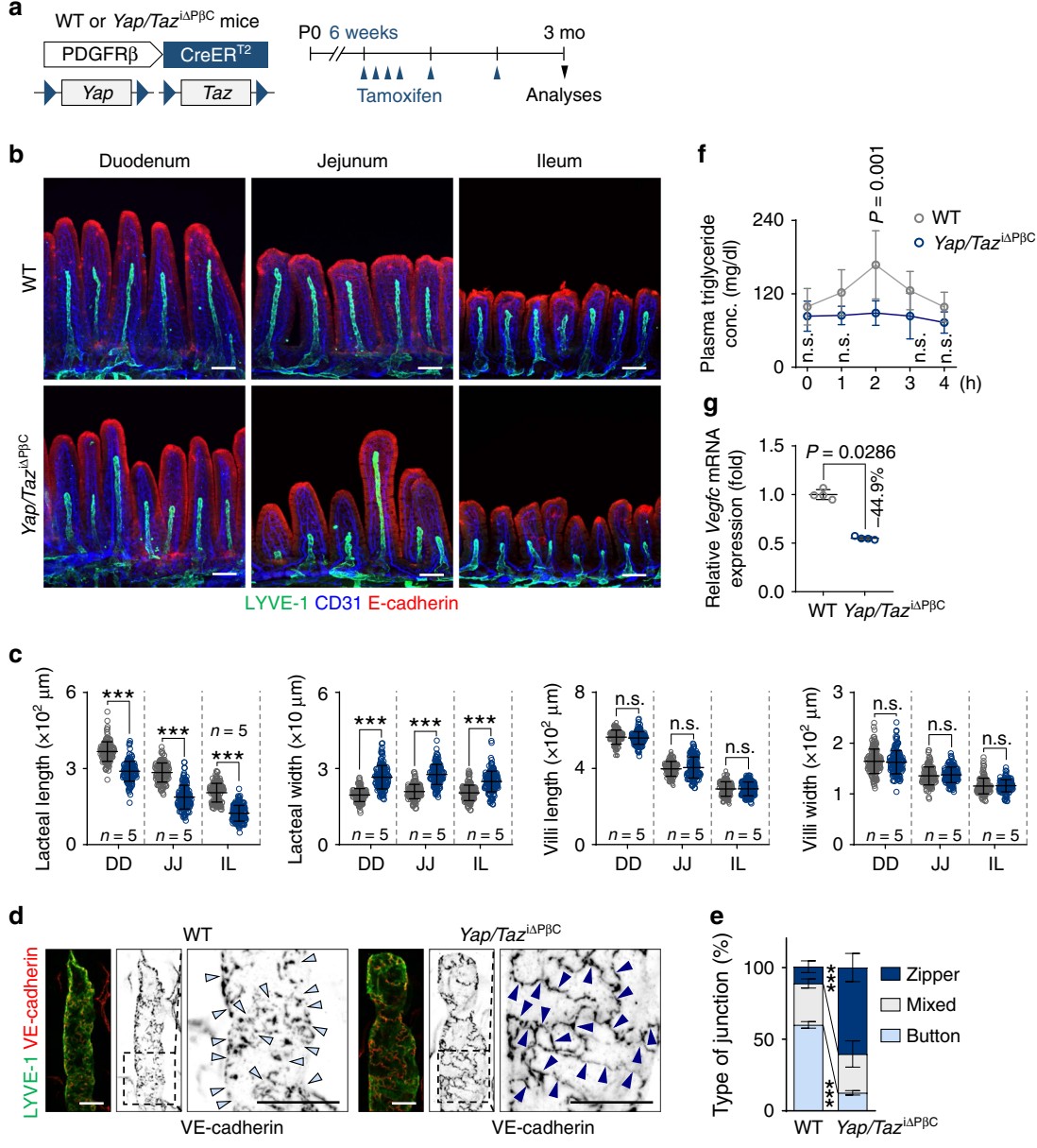

**Fig. 4 Regression of lacteal and attenuated *Vegfc* in intestinal villi of *Yap/Taz*[iΔPβC] mice. a** Diagram depicting the generation of *Yap/Taz*[iΔPβC] mouse and PDGFRβ[+] cell-specific depletion of *Yap/Taz* in small intestinal villi starting at 6 weeks and their analyses at 3 months after tamoxifen treatment. **b**, **c** Representative images of LYVE-1[+] lacteals and CD31[+] capillary plexus under the E-cadherin[+] intestinal epithelial cells and comparisons of the lacteal length, lacteal width, villi length, and villi width in duodenum (DD), jejunum (JJ), and ileum (IL) of the indicated part of small intestine in WT and *Yap/Taz*[iΔPβC] mice. Dots indicate values from 134~159 villi/group from three independent experiments using *n* = 5 mice/group. Horizontal bars indicate mean ± SD and *** *P* < 0.0001 versus WT by two-tailed Mann–Whitney *U* test. n.s., not significant. Scale bars, 100 μm. **d**, **e** Representative images and comparison of VE-cadherin[+] lymphatic endothelial cell (LEC) junctions of LYVE-1[+] lacteals in the villi of WT and *Yap/Taz*[iΔPβC] mice. Arrowheads indicate the button-type (WT) or zipper-type dominant junctional pattern (*Yap/Taz*[iΔPβC]). Horizontal bars of each colored segment represent mean ± SD of 5-10 villi/mouse and *n* = 5 mice/group pooled from three independent experiments. *** *P* < 0.0001 versus WT by two-way ANOVA with Holm–Sidak's multiple comparisons test. Scale bars, 20 μm. **f** Plasma triglyceride concentration (mg/dl) at the indicated time points after gavage of 200 μl of olive oil following fasting for 12 h in WT and *Yap/Taz*[iΔPβC] mice. Each dot indicates a value from *n* = 6 (WT) or *n* = 5 (*Yap/Taz*[iΔPβC]) mice pooled from three independent experiments. Horizontal bars indicate mean ± SD and *P* value versus WT by two-way ANOVA with Bonferroni's multiple comparisons test. **g** Comparison of *Vegfc* mRNA expression in the intestinal villi lysates of WT and *Yap/Taz*[iΔPβC] mice. Fold changes in mRNA expressions relative to the levels of WT mice are presented. Each dot indicates a value obtained from one mouse and *n* = 4 mice/group pooled from three independent experiments. Horizontal bars indicate mean ± SD and *P* value versus WT by two-tailed Mann–Whitney *U* test.

We further investigated whether YAP/TAZ are regulated by the mechanical and osmotic stress for VEGF-C secretion in isolated mouse IntSCs. We first disrupted actin-cytoskeleton by treatment with actin polymerization inhibitor (Latrunculin A) and non-muscle myosin inhibitor (Blebbistatin), because mechanical cues are propagated through the actin-cytoskeleton[29]. Perturbation of F-actin or non-muscle myosin inhibited YAP/TAZ nuclear localization and attenuated *Vegfc* transcription in isolated IntSCs (Fig. 5c, d). In parallel, matrix stiffness enforces different degrees of cell spreading[41]. Therefore,

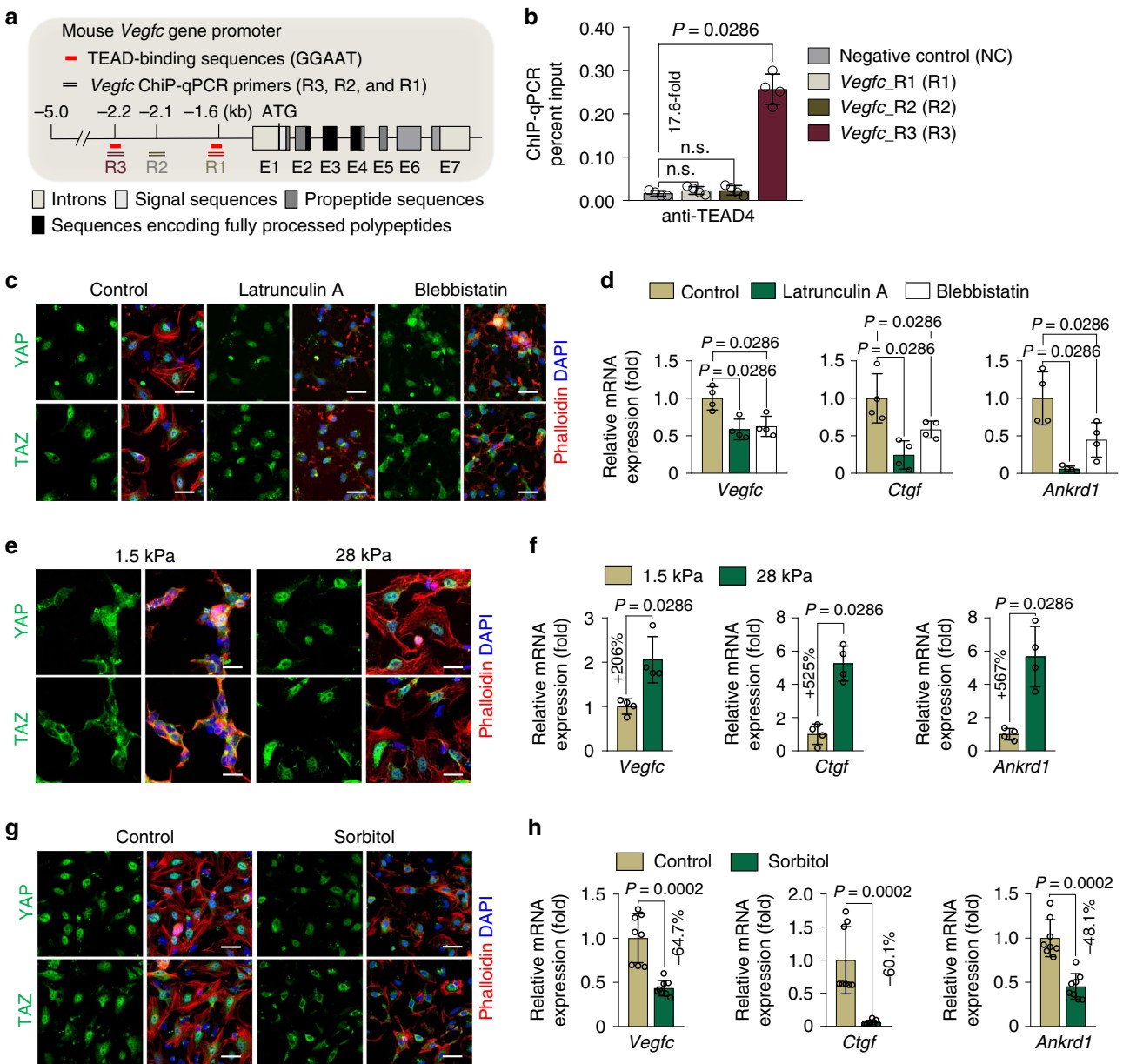

**Fig. 5 Mechanical and osmotic stress regulate YAP/TAZ and *Vegfc* in IntSCs. a** Diagram depicting the location of mouse *Vegfc* promoter element upstream of the transcription start site (ATG) relative to the *Vegfc* gene with presence and arrangement of consensus TEAD-binding sequences (GGAAT, red lines). ChIP-qPCR primers of *Vegfc* gene promoter around the TEAD-binding sequences are depicted as R1, R2 or R3. **b** ChIP-qPCR in MEFs indicating the endogenous interaction between TEAD and the *Vegfc* promoter located −2.2 kb upstream of the transcription start site of *Vegfc* gene. Dots indicate values from four independent experiments, and horizontal bars indicate mean ± SD. *P* value versus negative control (NC) by two-tailed Mann–Whitney *U* test. n.s., not significant. **c, d** Representative images of YAP and TAZ and comparison of mRNA expressions of *Vegfc*, *Ctgf*, and *Ankrd1* in isolated IntSCs after treatment with control (DMSO), latrunculin A (1 μM), or blebbistatin (50 μM) for 6 h. Similar results were observed in four independent experiments. Scale bars, 50 μm. Dots indicate data from four independent experiments and horizontal bars indicate mean ± SD. *P* value versus control by two-tailed Mann–Whitney *U* test. **e, f** Representative images of YAP and TAZ and comparison of mRNA expressions of *Vegfc*, *Ctgf* and *Ankrd1* in isolated IntSCs plated on 1.5 kPa and 28 kPa matrix. Similar results were observed in four independent experiments. Scale bars, 20 μm. Dots indicate data from four independent experiments and horizontal bars indicate mean ± SD. *P* value versus 1.5 kPa matrix by two-tailed Mann–Whitney *U* test. **g, h** Representative images of YAP and TAZ and comparison of mRNA expressions of *Vegfc*, *Ctgf* and *Ankrd1* in isolated IntSCs with or without 0.4 M sorbitol-induced osmotic stress for 3 h. Similar results were observed in four independent experiments. Scale bars, 50 μm. Dots indicate data from eight independent experiments and horizontal bars indicate mean ± SD. *P* value versus Control by two-tailed Mann–Whitney *U* test.

we seeded isolated IntSCs on soft (1.5 kPa) or stiff (28 kPa) hydrogels to induce differential mechanical stress. IntSCs seeded on stiff (28 kPa) matrix showed more spread phenotype than those seeded on soft (1.5 kPa) matrix (Fig. 5e). This mechanical cue induced nuclear localization of YAP/TAZ and enhanced mRNA expressions of *Vegfc* (~2.0-fold), *Ctgf* (~5.2-fold) and

*Ankrd1* (~5.6-fold) in IntSCs (Fig. 5e, f). Conversely, high osmolarity led to cytoplasmic localization of YAP/TAZ and attenuated expressions of *Vegfc* (64.7%), *Ctgf* (60.1%), *Ankrd1* (48.1%) in IntSCs (Fig. 5g, h). Furthermore, osmotic perturbation into the ex vivo culture of mouse small intestine and *in vivo*[42] promoted cytoplasmic translocations of YAP/TAZ in PDGFRβ+

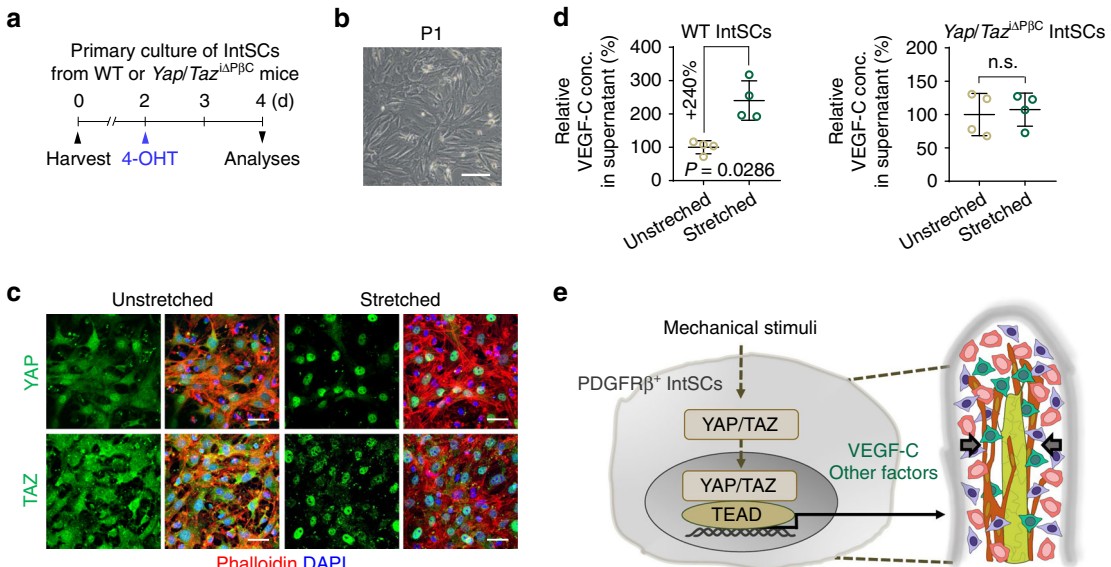

**Fig. 6 Mechanical stretching induces YAP/TAZ activation and VEGF-C secretion in IntSCs. a** Diagram for primary culture of IntSCs derived from WT or *Yap/Taz*^iΔPβC mice for 2 days and their analyses at 2 days after 100% EtOH or 5 µM of hydroxy-tamoxifen (4-OHT) treatment. **b** Phase-contrast image of freshly isolated primary mouse IntSCs from WT mice without passage (P1). Similar results were observed in four independent experiments. Scale bar, 200 µm. **c** Representative images of YAP and TAZ with phalloidin+ actin filaments in unstretched and stretched IntSCs. Similar findings were observed in four independent experiments. Scale bars, 50 µm. **d** Comparison of VEGF-C protein concentration in the culture medium of unstretched and stretched IntSCs derived from WT or *Yap/Taz*^iΔPβC mice. Dots indicate value from four independent experiments and horizontal bars indicate mean ± SD. *P* value versus unstretched by two-tailed Mann–Whitney *U* test. **e** Schematic images depicting upregulation and secretion of VEGF-C and other factors upon nuclear translocation and activation of YAP/TAZ and subsequent binding to TEAD after mechanical stimuli in PDGFRβ+ IntSCs.

IntSCs (Supplementary Fig. 13a–d). In concordance, while the protein level of VEGF-C substantially increased after mechanical stretch in the supernatant of primary IntSC culture medium from WT mice, VEGF-C level was not significantly altered in the *Yap/Taz* depleted primary IntSCs (Fig. 6a–d and Supplementary Fig. 14). Collectively, our data imply that YAP/TAZ in PDGFRβ+ IntSCs contribute to maintain lacteal integrity and function by dynamically regulating VEGF-C expression in response to mechanical cues and osmotic stress (Fig. 6e).

**Single-cell RNA sequencing of PDGFRβ+ IntSCs unravels their heterogeneity and identifies distinct subsets of fibroblasts that secrete VEGF-C.** VEGF-C was suggested to be secreted from the enteric SMCs, which are surrounding the lacteal[5]. As we observed expression of PDGFRβ in enteric SMCs of the intestinal villi using *tdTomato*^rPβC mice (Supplementary Fig. 6c), we investigated whether the lacteal phenotypes of *Lats1/2*^iΔPβC mice are induced by the VEGF-C secreted from the surrounding PDGFRβ+ SMCs or PDGFRβ+ non-SMCs. We generated *tdTomato*^rSMC by crossing *Myh11*-Cre-ER^T2 mouse with the *Rosa26-tdTomato* reporter mouse and confirmed co-distribution of αSMA staining on Myh11+ SMCs in the small intestinal villi (Supplementary Fig. 15a, b). By generating *Lats1/2*^iΔSMC mice by crossing *Myh11*-Cre-ER^T2 mouse with *Lats1*^fl/fl/*Lats2*^fl/fl mouse, we found that lacteal phenotypes were not so remarkable compared with those of *Lats1/2*^iΔPβC mice; still, the alignment and shape of SMCs along the villi were altered and villi structure and capillary plexus were shrunk overall (Supplementary Fig. 15c–f). Notably, the increase in the number of lacteal Prox1+ LECs, aberrant lacteal sprouting, and hyperbranching that were observed in *Lats1/2*^iΔPβC mice were not fully recapitulated in *Lats1/2*^iΔSMC mice (Supplementary Fig. 15g, h). These findings suggest that YAP/TAZ hyper-activation in PDGFRβ+ enteric SMCs alone endows a certain but minor effect on the aberrant lacteal phenotype compared

with *Lats1/2*^iΔPβC mice, which are targeting both PDGFRβ+ SMCs and PDGFRβ+ non-SMCs.

To delineate the detailed source of VEGF-C among the heterogeneous PDGFRβ+ IntSCs, we performed droplet-based single-cell mRNA sequencing of PDGFRβ+ IntSCs isolated from adult *tdTomato*^rPβC mice (Fig. 7a, b). We retained profiles of 7906 cells with a median of 3,672 detected genes per cell. Unsupervised clustering of the PDGFRβ+ IntSCs revealed seven distinct cell clusters visualized with uniform manifold approximation and projection (UMAP); while all cells expressed high levels of *Pdgfrb* (Fig. 7c).

When we examined differentially expressed genes to characterize each cluster, seven distinct cell clusters exhibited highly cluster-specific gene expression profiles (Fig. 7d, e). Importantly, we were able to identify five distinct clusters of PDGFRβ+ fibroblasts, of which three have not been described before (Fig. 7c–e). Intestinal villi fibroblast (vFB) 1 expressed high levels of immediate early genes, such as *Junb* and *Fosb*. Although expression of immediate early genes could arise as an artifact due to cell dissociation procedures, PDGFRβ+Fosb+ cells were detected in vivo regardless of cellular dissociation. vFB2 shared a similar transcriptomic profile with vFB1 but without the expression of immediate early genes (Fig. 7d, e). Of note, matrix metalloproteinase 10 (*Mmp10*) and four and a half LIM domains protein 2 (*Fhl2*) genes were exclusively expressed in vFB3, suggesting a role of this cluster in extracellular matrix remodeling (Fig. 7d, e). FB4 discretely expressed atypical chemokine receptor 4 (*Ackr4*) (Fig. 7d, e), which has been reported to be expressed in submucosal fibroblasts[43]. Finally, FB5 exclusively expressed high levels of SRY-Box 6 (*Sox6*) and bone morphogenic protein 7 (*Bmp7*), which have been shown to be expressed in fibroblasts lining the epithelium of the small intestinal villi[44]. Of the remaining 2 clusters, one expressed the hallmarks of enteric SMCs including calponin 1 (*Cnn1*) and actin gamma 2 (*Actg2*), while the other cluster expressed vitronectin (*Vtn*) which marks mural cells (Fig. 7e).

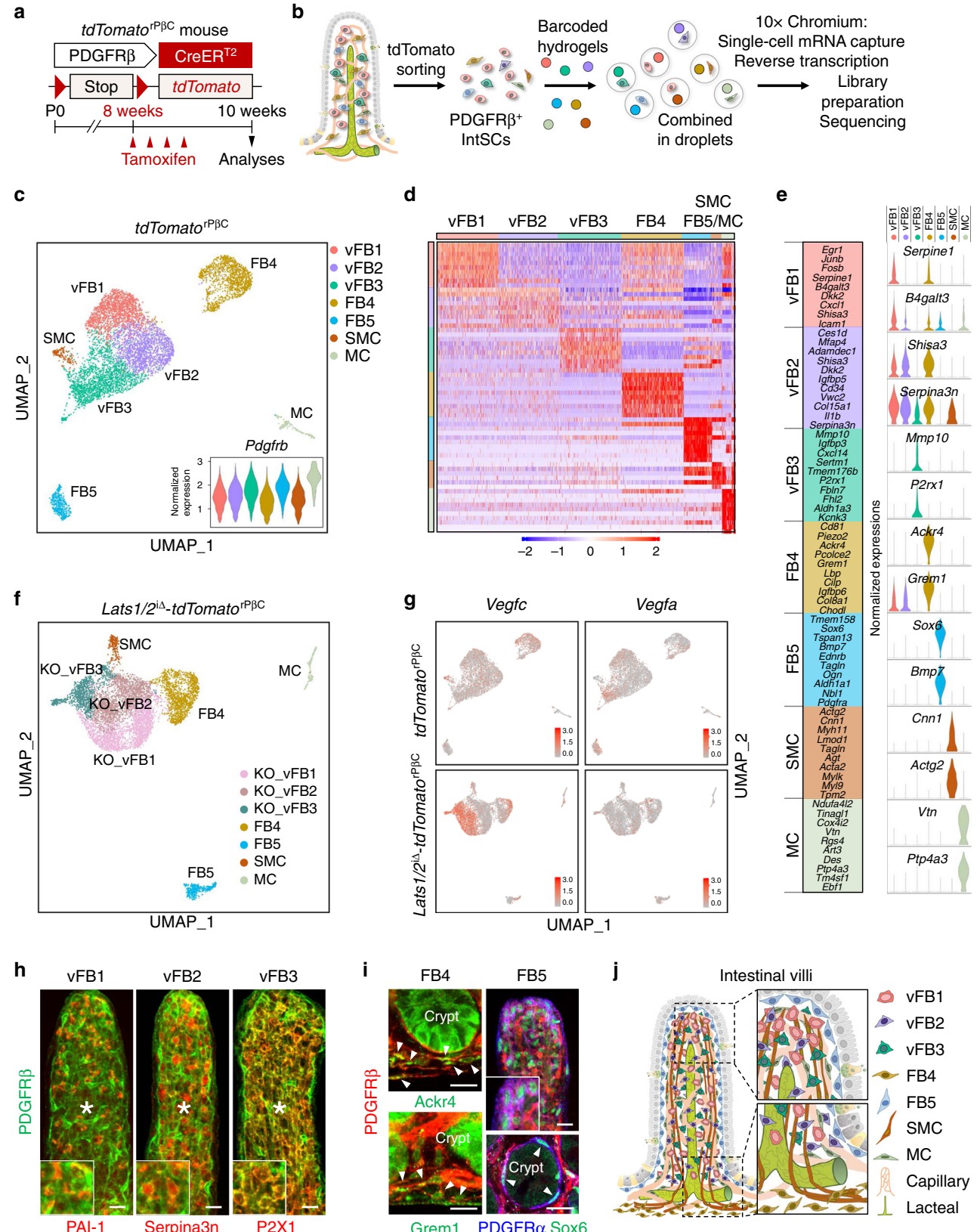

We also generated a single cell library from *Lats1/2*[iΔ]-*tdTomato*[rPβC] mice (KO_) to delineate the villi PDGFRβ+ cells that secrete VEGF-C upon YAP/TAZ hyperactivation, and found seven distinct PDGFRβ+ IntSC clusters as observed in *tdTomato*[rPβC] mice (Fig. 7f and Supplementary Fig. 16a, b). For comparative analysis on the transcriptomic alterations among clusters, we integrated the datasets of *tdTomato*[rPβC] and *Lats1/2*[iΔ]-*tdTomato*[rPβC] mice, each composed of two mice per group, by multiple dataset integration methods in Seurat and performed unsupervised clustering (Supplementary Fig. 16c). When we

**Fig. 7 Single-cell RNA sequencing reveals heterogeneity and distinct subsets of fibroblasts. a** Diagram depicting the generation of $tdTomato^{rP\beta C}$ mouse and PDGFRβ+ cell-specific expression of tdTomato from 8-week old and their analyses at 10-week old. **b** Illustration of droplet-based single-cell RNA sequencing (scRNA-seq) of PDGFRβ+tdTomato+ IntSCs sorted from the small intestine of $tdTomato^{rP\beta C}$ mice. **c** Visualization of unsupervised clustering of 7 distinct PDGFRβ+ IntSC clusters by Uniform manifold approximation and projection (UMAP) in the small intestine of $tdTomato^{rP\beta C}$ mice. vFB, intestinal villi fibroblast; FB, fibroblast; SMC, smooth muscle cell; MC, mural cell. **d** Heatmap displaying the scaled expression patterns of top ten differentially expressed genes for random sampled cells (maximum thousand cells) for each indicated clusters. **e** List of representative marker genes of each of the seven PDGFRβ+ IntSC clusters (left) and violin plots (right) showing the expression of top-ranking marker genes for each cluster. Log normalized read counts as y-axis (normalized expression). **f** UMAP visualization of unsupervised clustering of seven distinct PDGFRβ+ IntSC clusters in the small intestine of $Lats1/2^{i\Delta}$-$tdTomato^{rP\beta C}$ mice. **g** Gene expression levels of Vegfc and Vegfa in $tdTomato^{rP\beta C}$ and $Lats1/2^{i\Delta}$-$tdTomato^{rP\beta C}$ mice projected on UMAP plot. Note that specific subsets of PDGFRβ+ IntSCs (vFB1-3) in the small intestine of $Lats1/2^{i\Delta}$-$tdTomato^{rP\beta C}$ mice show higher expression of Vegfc compared with those of $tdTomato^{rP\beta C}$ mice. **h, i** Representative images of PDGFRβ+ IntSCs in WT mouse reveal expressions of each fibroblast-specific markers: PAI-1+ vFB1, Serpina3n+ vFB2, P2X1+ vFB3, Ackr4+ or Grem1+ FB4, and PDGFRα+ Sox6+ FB5. Each white box in the lower left corner is a magnified view. White asterisks indicate lacteals and white arrowheads indicate each fibroblast cluster-specific cell type stained with the indicated marker. Similar findings were observed in $n = 4$ mice from two independent experiments. Scale bars, 20 μm. **j** Schematic images depicting the anatomic distribution of indicated markers for vFB1-3, FB4, FB5, SMC, MC, capillary plexus, and lacteal in intestinal villi of adult WT mouse. vFB1-3 are uniformly distributed around the lacteal, whereas FB4 is mainly located in the submucosal area and FB5 is mostly placed under the intestinal epithelium. Black dashed boxes are magnified in the right panels.

measured the correlation of clusters from both datasets, each of the seven distinct PDGFRβ+ IntSCs demonstrated highly correlative expression of marker genes, with no significant changes in cell type abundances (Supplementary Fig. 16c, d). Of special note, KO_vFB1-3 populations demonstrated robust upregulation of Vegfc, but not Vegfa, upon YAP/TAZ hyper-activation regardless of the duration after the tamoxifen delivery (Fig. 7g and Supplementary Fig. 16e). Moreover, gene ontology suggested possible discrete functions in vFB1-3 (Supplementary Fig. 16f). These results indicate that distinct subsets of intestinal fibroblasts (vFB1-3) upregulate Vegfc upon YAP/TAZ hyperactivation.

To define the location of these newly identified subsets of fibroblasts, we examined and validated the expressions of markers that are specific to each cluster (Fig. 7h, i). Of note, vFB1-3 were distributed in close proximity to the lacteal, while Ackr4+ FB4 and Sox6+ FB5 were localized around the submucosal and subepithelial regions, respectively (Fig. 7h, i), as previously reported[12,43]. vFB1-3 were distinguished by specific markers: P2X1 was exclusively expressed by vFB3 but not by vFB1-2; Fosb was specifically expressed by vFB1 but not by vFB2 (Fig. 8a, b). In addition, through single-molecule in situ hybridization (smFISH) of Vegfc, we observed that the expression of Vegfc is upregulated in $Lats1/2^{i\Delta}$-$tdTomato^{rP\beta C}$ mice compared with $tdTomato^{rP\beta C}$ mice (Fig. 8c). Moreover, we performed smFISH of Vegfc with immunofluorescence staining of vFB1-3-specific markers. Vegfc expression increased in vFB1-2 (Shisa3+PDGFRβ+ cells) and vFB3 (P2X1+PDGFRβ+ cells) in $Lats1/2^{i\Delta P\beta C}$ mice compared with WT mice (Fig. 8d). Taken together, our data unravel the heterogeneity and new subtypes of IntSCs (Fig. 7j) and suggest that vFB1-3 maintain lacteal integrity and function by forming a YAP/TAZ-induced VEGF-C secreting niche.

## Discussion

Our most intriguing finding is that intestinal villi fibroblasts regulate lacteal integrity through YAP/TAZ-induced secretion of VEGF-C upon mechanical cues. Taking advantage of single-cell RNA sequencing and detailed 3D analysis of IntSCs, we further reveal the heterogeneity of villi fibroblasts and characterize their distinct transcriptomic features with unique distributions in the intestinal villi.

The molecular signature and functional properties of lymphatic vessels are highly dependent on the stroma where they reside and VEGF-C/VEGFR3 signaling pathway plays major role in lymphangiogenesis[2,3,45]. SMCs may be an important source of VEGF-C, which is required to maintain lymphatic vessel architecture in the intestine[5]. Nevertheless, extrinsic factors governing lacteal maintenance and function require better delineation. Hypersprouting and hyperbranching phenotypes were strikingly found in lacteals after YAP/TAZ hyperactivation in PDGFRβ+ IntSCs. This was due to the paracrine activation of VEGFR3 on neighboring LECs, triggered by high VEGF-C secretion from PDGFRβ+ fibroblast subpopulations. A recent report suggested that lacteal LEC junction zippering and disassembly of cytoskeletal VE-cadherin anchors prevent dietary fat uptake[4–7]. In our experiments, we observed zippering of lacteal junctions and impaired dietary fat uptake with increased or decreased VEGF-C after hyperactivation or depletion of YAP/TAZ, respectively, in PDGFRβ+ IntSCs. Thus, our results indicate that VEGF-C/VEGFR3 signaling is important for maintaining the button-like lacteal junction for proper dietary fat uptake, together with VEGF-A/VEGFR2 signaling[4–7].

We also demonstrated the importance of mechanical cues in regulating nucleocytoplasmic shuttling of YAP/TAZ in fibroblasts, for VEGF-C secretion possibly along with other growth factors and cytokines. Owing to their unique circumstances, intestinal lymphatic capillaries as well as surrounding IntSCs must withstand high levels of repetitive mechanical stress, extreme osmotic variations, and commensal bacteria[1–3]. To support lacteal integrity and function in such environments, some mechanisms must exist for IntSCs to adapt and respond to various types of stress. We showed that YAP/TAZ respond to mechanical or osmotic stress by promoting or attenuating VEGF-C secretion in fibroblasts, enabling them to sense and adapt to high mechanical forces and osmotic stress. Collectively, these findings point to an important link between YAP/TAZ mechanotransduction and biological outcomes in the intestine, likely dictated by the physiology of the gastrointestinal tract.

Our analysis uncovering the five transcriptionally distinct subsets of intestinal fibroblasts highlights the heterogeneity and novel role of intestinal fibroblasts in maintaining lacteal homeostasis. Accumulating evidence continues to indicate the importance and diverse tissue-specific roles of perivascular fibroblasts in normal physiology, as well as during tissue remodeling after injury[14,16,46]. In agreement, fibroblasts are not only a mechanical scaffold for capillaries but also implicated as an important source of endothelial growth factors that mediate capillary morphogenesis and vessel maintenance[47]. Therefore, the intimate crosstalk between fibroblasts and capillaries seems necessary to support the homeostasis of the organ they pervade in a tissue-specific manner[47]. However, little is understood regarding IntSC structural and functional heterogeneity and their interaction. We have

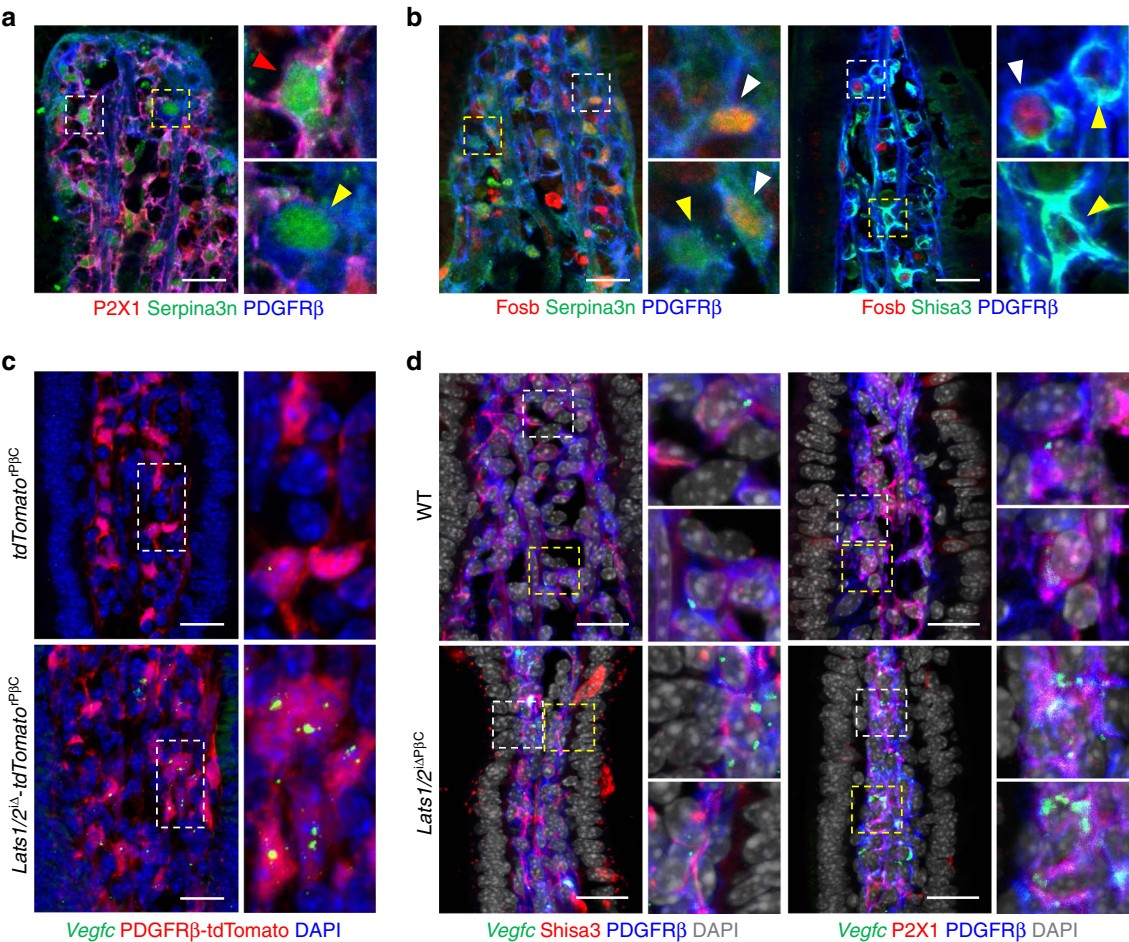

**Fig. 8 vFB1-3 are distinct cell types secreting VEGF-C. a** Representative images of vFB3 (P2X1+Serpina3n+PDGFRβ+) and vFB1-2 (P2X1-Serpina3n+PDGFRβ+) in WT mouse. Similar findings were observed in $n = 4$ mice from two independent experiments. White and yellow dotted boxes are magnified in the upper right and lower right panels, respectively. Red and yellow arrowheads indicate vFB3 and vFB1-2, respectively. Scale bar, 20 µm. **b** Representative images of vFB1 (Serpina3n+Fosb+Shisa3+PDGFRβ+) and vFB2 (Serpina3n+Fosb-Shisa3+PDGFRβ+) in WT mouse. Similar findings were observed in $n = 4$ mice from two independent experiments. White and yellow dotted boxes are magnified in the upper right and lower right panels, respectively, for each image. White and yellow arrowheads indicate vFB1 and vFB2, respectively. Scale bars, 20 µm. **c** Representative images of *Vegfc* single-molecule fluorescence in situ hybridization (smFISH) in intestinal villi of *tdTomato*rPβC and *Lats1/2*iΔ- *tdTomato*rPβC mice. Similar findings were observed in $n = 4$ mice/group from two independent experiments. White dotted boxes are magnified in the right panel. Scale bars, 20 µm. **d** Representative images of *Vegfc* smFISH in vFB1-3 of WT and *Lats1/2*iΔ- *tdTomato*rPβC mice. Similar findings were observed in $n = 4$ mice/group from two independent experiments. White and yellow dotted boxes are magnified in the upper right and lower right panels, respectively, for each image. Scale bars, 20 µm.

uncovered the molecular identity of intestinal fibroblasts and revealed their unique anatomic locations, which reflect their discrete functions. Of note, ACKR4+ fibroblasts in the villi stroma, equivalent to the FB4 subtype here, reside exclusively in the intestinal submucosa and are suggested to control leukocyte migration[43]. Indeed, FOXL1+ fibroblasts in the intestinal sub-epithelial compartment, which correspond to the FB5 subtype in this work, have been suggested as an essential source of Wnt protein for gut epithelial renewal[12].

Most importantly, we identified novel distinct subsets of fibroblasts (FB1–3), which were distributed adjacent to the lacteals and are potential regulators of lacteal integrity and function by YAP/TAZ-induced VEGF-C secretion. We also have suggested cell type relationships among FB1–3 and their possible functions. Many evidences suggest the importance and multiple roles of intestinal fibroblasts and mesenchymal stem cells in maintaining intestinal homeostasis and in immunity and disease[48,49]. However, more extensive work is needed to understand the

phenotypic and functional heterogeneity within IntSCs. Our study opens the way for using Cre-expressing reporter and fate-mapping mouse lines to reveal lineage relationships and their functions of novel IntSC subtypes. This work will contribute to a better characterization of populations associated with maintaining intestinal homeostasis or mediating intestinal dysfunction.

Taken together, our findings advance understanding of IntSCs and reveal a previously unknown role for villi fibroblasts as essential regulators of lacteal integrity. We highlight unique features of organ-specific lymphatic capillaries that coordinates with the surrounding fibroblasts in the intestinal stroma to accomplish functional interplay between YAP/TAZ and lymphatic maintenance. However, the resulting phenotypes of genetically modified mice and the experimental tools used to induce mechanical or osmotic stress are more extreme than those observed in physiological conditions. Nevertheless, our results reinforce the concept that the paracrine molecules secreted from adjacent

fibroblasts are the major regulators of lymphangiogenesis and lymphatic remodeling. These findings fill in parts of the novel dynamic regulatory mechanism of lacteals and offer clues to the process of maintaining intestinal homeostasis.

## Methods

**Experimental animals**. All animal care and experimental procedures were complied with all ethical regulations for animal research and testing under the approval from the Institutional Animal Care and Use Committee (No. KA2016-12) of Korea Advanced Institute of Science and Technology (KAIST). *Pdgfrb*-Cre-ER[T2] [30], *Lats1*[fl/fl]/*Lats2*[fl/fl][31,32] and *Yap*[fl/fl]/*Taz*[fl/fl] [38] mice were transferred, established, and bred in specific pathogen-free (SPF) animal facilities at KAIST. C57BL/6 J, R26-tdTomato and *Myh11*-Cre-ER[T2] mice were purchased from the Jackson Laboratory. All mice were maintained in the C57BL/6 background and fed with free access to a standard diet (PMI LabDiet) and water. In order to induce Cre activity in the Cre-ER[T2] mice, 2 mg of tamoxifen (Sigma-Aldrich) dissolved in corn oil (Sigma-Aldrich) was injected intraperitoneally (i.p.) at indicated time points for each experiment. Cre-ER[T2] negative but flox/flox-positive mice among the littermates were defined as control (WT) mice for each experiment. Mice were anesthetized with i.p. injection of a combination of anesthetics (80 mg/kg ketamine and 12 mg/kg of xylazine) before being sacrificed.

**Histological analyses**. For whole-mount staining of small intestine, transcardial perfusion was performed with 2% paraformaldehyde (PFA) after anesthesia. Small intestine was harvested and cut longitudinally to expose the lumen. After several washes with PBS, intestines were pinned on silicon plates. Samples were then post-fixed at 4 °C in 4% PFA for 2 h. Samples were washed with PBS several times and were subsequently dehydrated with 10% sucrose in PBS for 2 h and with 20% sucrose, 10% glycerol in PBS overnight. Ear skin, trachea and diaphragm were harvested without transcardial perfusion and fixed in 4% PFA for 1 h at 4 °C. Samples were washed with PBS several times before blocking. After blocking with 5% goat or donkey serum in 0.5% Triton-X 100 in PBS (PBST) for 1 h, samples were incubated with the indicated primary antibodies diluted in the blocking solution at 4 °C overnight. After several washes with PBST, the samples were incubated for 2 h at room temperature (RT) with the indicated fluorocrome-conjugated secondary antibodies diluted in the blocking buffer. The samples were washed with PBST and nuclei were stained with DAPI (Invitrogen). After washing with PBS, samples were mounted with Vecta-shield (Vector Laboratories). Images were obtained using Zeiss LSM 800 or LSM 880 confocal microscope (Carl Zeiss).

The following primary and secondary antibodies were used in the immunostaining: anti-LYVE-1 (rabbit polyclonal, 11-034, Angiobio, 1:400); anti-CD31 (rat monoclonal, 557355, BD Biosciences, 1:400); anti-CD31 (hamster monoclonal, MAB1398Z, Merck, 1:400); anti-E-cadherin (goat polyclonal, AF748, R&D, 1:200); anti-Prox1 (goat polyclonal, AF2727, R&D, 1:400); anti-Prox1 (rabbit polyclonal, 102-PA32AG, ReliaTech, 1:400); anti-VE-cadherin (goat polyclonal, AF1002, R&D, 1:200); anti-PDGFRβ (rat monoclonal, ab91066, Abcam, 1:200); anti-PAI-1 (mouse monoclonal, sc-5297, Santa Cruz, 1:100); anti-Serpina3n (Goat polyclonal, AF4709, R&D, 1:200); anti-Fosb (rabbit monoclonal, 2251, Cell Signaling Technology, 1:400); anti-Shisa3 (rabbit polyclonal, TA320118, Origene, 1:400); anti-P2X1 (rabbit polyclonal, APR-001, Alomone labs, 1:800); anti-Ackr4 (rabbit polyclonal, SAB4502137, Sigma-Aldrich, 1:200); anti-Grem1 (goat polyclonal, AF956, R&D, 1:200); anti-Sox6 (rabbit polyclonal, ab30455, Abcam, 1:400); anti-PDGFRα (goat polyclonal, AF1062, R&D, 1:200); anti-YAP (rabbit monoclonal, 14074, Cell signaling, 1:200); anti-TAZ (rabbit polyclonal, HPA007415, Sigma-Aldrich, 1:200); anti-αSMA, Fluorescein Isothiocyanate (FITC)-conjugated (mouse monoclonal, F3777, Sigma-Aldrich, 1:1000); anti-VEGFR3 (goat polyclonal, AF743, R&D, 1:200); anti-VEGFR2 (goat polyclonal, AF644, R&D, 1:200); anti-PGP9.5 (rabbit monoclonal, 13179, Cell signaling, 1:400); anti-F4/80, FITC-conjugated (rat monoclonal, 1231007, Biolegend, 1:200); anti-CD3 (hamster monoclonal, 553058, BD Biosciences, 1:1000); anti-Desmin (rabbit polyclonal, AB907, Millipore, 1:400); and Alexa Fluor 488-, Alexa Fluor 594-, Alexa Fluor 647-conjugated anti-rabbit, anti-rat, anti-goat, anti-hamster secondary antibodies (diluted at a ratio of 1:1000) were purchased from Jackson ImmunoResearch. All the antibodies used in our study were validated for the species and applications.

**Single-molecule fluorescence in situ hybridization (smFISH)**. For smFISH of small intestine, transcardial perfusion was performed with 2% PFA after anesthesia. Samples were post-fixed at 4 °C in 4% PFA for 2 h. Samples were washed with PBS several times, moved to 30% sucrose, and incubated overnight. The samples embedded in OCT were sectioned, and we performed the smFISH according to the manufacturer's protocol that is described in the RNAscope Fluorescent Multiplex Assay kit (320850, ACDBio). The RNA scope Probe (ACDBio)-Mm-*vegfc* (492701-C2) was used for smFISH. The following primary and secondary antibodies were used in the immunostaining with smFISH: anti-PDGFRβ (rat monoclonal, ab91066, Abcam); anti-Shisa3 (rabbit polyclonal, TA320118, Origene); anti-P2X1 (rabbit polyclonal, APR-001, Alomone labs) and Alexa Fluor 488-, Alexa Fluor 594-, Alexa Fluor 647-conjugated secondary antibodies were purchased from

Jackson ImmunoResearch. Images were obtained using Zeiss LSM 800 or LSM 880 confocal microscope (Carl Zeiss).

**EdU incorporation assay for proliferating LECs**. To detect proliferating LECs in lacteal, 5 mg of 5-ethynyl-2′-deoxyuridine (EdU, A10044, Invitrogen) was dissolved in 1 ml of Milli-Q water as a stock solution. Then, 200 μl of the stock solution per mice was injected i.p. every other day for a week before analysis. Small intestine was isolated and processed as described above. EdU-incorporated cells were detected with the Click-iT EdU Alexa Fluor-488 Assay Kit (Invitrogen) according to the manufacturer's protocol.

**Electron microscopy**. To capture ultrastructure electron microscopic images of lacteal, small intestine was sectioned after transcardial perfusion with 4% PFA and 0.25% glutaraldehyde in 0.1 M phosphate buffer (pH 7.4). Samples were then fixed overnight in 2.5% glutaraldehyde, post-fixed with 1% osmium tetroxide, and dehydrated with a series of increasing ethanol concentrations followed by resin embedding. 70-nm ultrathin lacteal sections were obtained using an ultra-microtome (UltraCut-UCT, Leica), which were then collected on copper grids. Samples were imaged with transmission EM (Tecnai G2 Spirit Twin, FEI) at 120 kV after staining with 2% uranyl acetate and lead citrate.

**Intravital imaging of lipid clearance from lamina propria**. Mice were anesthetized following food removal for 12 h before the procedure. Intravital imaging was performed as we previously described[34]. Briefly, BODIPY lipid probes (0.1 mg/ml, Thermo Fisher) were dissolved in 2.5% DMSO solution (Sigma-Aldrich). Anti-LYVE-1 (rat monoclonal, 223322, R&D) conjugated with Alexa Fluor 647 (Invitrogen) antibody (0.75 mg/kg) was intravenously injected at 12 h before the imaging for fluorescent labeling of lacteals. Then, the proximal jejunum was exteriorized in an imaging chamber where the temperature was maintained at 37 °C during the imaging. After surgical opening of the intestinal lumen 1.5 cm along the anti-mesenteric border, a cover glass was placed on the exposed lumen to obtain a villi view. The intravital imaging was done over three sessions. The villi were first imaged before the BODIPY-FA supply at 0 min. Then, 30 μl of BODIPY lipid probes were supplied once before the second imaging session to observe the initial BODIPY lipid absorption at 1 min. Next, the BODIPY lipids were supplied three times with 2 min intervals before the third imaging session at 26 min, and BODIPY-FA clearing through lacteals was analyzed at 36 min and 46 min. A home-built video-rate laser-scanning confocal microscope was used[34]. Two continuous-wave lasers emitting at 488 nm (MLD, Coherent) and 640 nm (Cube, Coherent) were used as excitation sources for fluorescence imaging. Two bandpass filters (FF01-525/50 and FF01-685/40, Semrock) were used for detection of fluorescent signals. Axial resolution below 4 μm was acquired with 100 μm pinhole and 60x objective lens (LUMFLN, water immersion, NA 1.1, Olympus). Images (512 × 512 pixels) were obtained at a frame rate of 30 Hz. For the signal-to-noise ratio enhancement of the image, the noise over 90 frames after post-processing the real-time images (30 frames/sec) were averaged by removing the motion artifact generated from peristalsis with a custom-written MATLAB program. ImageJ software (NIH) was used for measuring the average fluorescent intensity in the lamina propria.

**Lipid absorption test**. Following food removal for 12 h, 200 μl of olive oil (Sigma-Aldrich) was delivered by oral gavage for plasma triglyceride measurement and Oil Red O staining. Blood was sampled via the tail vein into the blood collection tube containing heparin at 0, 1, 2, 3, and 4 h after olive oil administration. Plasma triglyceride concentration was measured using a VetTest Chemistry analyzer (IDEXX Lab). Small intestine was stained with Oil red O at 12 h after the olive oil gavage following the manufacturer's protocol of the Oil Red O Staining Kit (MAK194, Sigma-Aldrich). High fat diet (HFD) (60% kcal fat, Research Diets, D12492) feeding started at 12-week old and continued for 8 weeks for plasma triglyceride measurement and Oil Red O staining. Blood was sampled via the tail vein into the blood collection tube containing heparin after 6 h fasting. For Oil red O staining of liver sections, liver harvested, fixed in 4% PFA overnight at 4 °C, were cut to 100 μm vibratome sections (Leica, VT1200S), were stained with Oil red O.

**Transduction of AAV**. Indicated mice received a single i.p. dose ($10^{12}$ viral particles in 150–200 μl) of a recombinant AAV encoding the ligand binding domains 1–4 of VEGFR3 fused to the IgG Fc domain (AAV–mVEGFR3$_{1-4}$-Ig), and control mice received the same dose of AAV encoding the domains 4–7 of VEGFR3, which do not bind VEGF-C or VEGF-D (AAV–mVEGFR3$_{4-7}$-Ig), fused to the IgG Fc domain, as previously described[37]. AAV–mVEGFR3$_{1-4}$-Ig and AAV–mVEGFR3$_{4-7}$-Ig were detected in serum by immunoblotting analysis (anti-VEGFR3 antibody; goat polyclonal, AF743, R&D) of 0.5 μl of serum samples collected at the same day of the intestinal analysis.

**Treatment of antibody**. For VEGFR2 blockade, we used a VEGFR2 neutralizing antibody (DC101). A hybridoma cell line that produces DC101 was purchased from American Type Culture Collection (ATCC). Hybridoma cells were grown in

serum-free medium. Recombinant proteins in the supernatants were purified by column chromatography with Protein A agarose gel (Oncogene). After purification, the recombinant proteins were quantified using the Bradford assay and confirmed by Coomassie blue staining after sodium dodecyl sulfate (SDS)–polyacrylamide gel electrophoresis (PAGE). DC101 (50 mg/kg) or an equal amount of the control antibody (IgG-Fc, SAB3700546, Sigma-Aldrich) was i.p. injected into the indicated mice every 2 days for 2 weeks.

**Intestinal LEC enrichment and IntSC isolation from the small intestine**. The serosa, muscle layer, and mesenteric adipose tissues were removed from the small intestine. After opening gut fragments longitudinally and washing with PBS, samples were cut into 2 cm pieces, and Peyer's patches were removed. Samples were washed with PBS and incubated with 10 mM EDTA with calcium- and magnesium-free DMEM (Gibco) on ice for 20 min. Tissues were vortexed with calcium-free PBS until obtaining a clear supernatant that was devoid of epithelial cells. Sample pieces were cut further into 1 mm fragments and dissociated with dissociation buffer containing 2 mg/ml collagenase II (Worthington), 1 mg/ml Dispase (Gibco), and 1 U/ml DNase (Invitrogen) in DMEM at 37 °C for 30 min. To help dissociation, tissues pieces were gently pipetted up and down every 10 min. The samples were then filtered through a 70 μm strainer to mechanically dis-aggregate the remaining intestinal fragments. After collecting the supernatants, an equal volume of DMEM containing 10% fetal bovine serum (FBS) was added. After centrifugation, cells were resuspended in PBS. To enrich the stromal cell fraction and LECs, hematopoietic cells and epithelial cells were depleted using AutoMACS (Miltenyi) after incubation for 15 min on ice with anti-CD45 and anti-CD326 Microbeads (Miltenyi). For IntSC isolation, anti-CD31 Microbeads (Miltenyi) were added to deplete endothelial cells in addition to the anti-CD45 and anti-CD326 Microbeads.

**Primary culture of mouse IntSCs**. Following the isolation of IntSCs, cells were cultured with DMEM/F12 containing 10% FBS on a tissue culture plate for 24 h or 2 days. For drug treatments and immunofluorescence staining, 50,000 cells per well were plated on 4-well Nunc Lab-Tek II chamber slides (Sigma-Aldrich). Drug concentrations are described in the indicated figure legends and treatments lasted 6 h for immunofluorescence and gene expression analysis. A 35-mm imaging dish with an Elastically Supported Surface (Ibidi) was used for IntSC culture on soft (1.5 kPa) and stiff (28 kPa) matrices for 24 h. For induction of cre activity in primary cultured IntSCs derived from WT or Yap/Taz$^{iΔFRC}$ mice, cells were treated with 5 μM of 4-hydroxy-tamoxifen (4-OHT) in 100% ethanol (EtOH) or 100% EtOH alone for 2 days as a control. Culture-expanded monolayer of IntSCs that were validated as PDGFRβ$^+$ were used for experiments.

**Flow cytometry and cell sorting**. To sort PDGFRβ$^+$ IntSCs or intestinal LECs, the enriched fractions went on RBC lysis by suspension in ACK lysis buffer (Gibco) for 5 min at RT. After blocking Fcγ receptors with mouse anti-CD16/CD32 (553141, BD Bioscience), cells were incubated for 15 min with indicated antibodies in FACS buffer (2% FBS in PBS). After several washes, cells were analyzed by FACS Canto II (BD Biosciences) and the acquired data were further evaluated in FlowJo software (Treestar). Cell sorting was performed with FACS Aria Fusion (Beckton Dickinson). Dead cells were excluded using DAPI (Sigma-Aldrich) staining and cell doublets were systematically excluded. Fluorescence intensity is expressed in arbitrary units on a logarithmic scale, and forward scatter and side scatter are represented on a linear scale. The following antibodies were used for flow cyto-metry: FITC anti-mouse CD45 (11-0451-85, rat monoclonal, Biolegend); FITC anti-mouse TER-119 (11-5921-85, rat monoclonal, Biolegend); APC anti-mouse CD31 (551262, rat monoclonal, BD Bioscience); and PE/Cy7 anti-mouse Podo-planin (127412, syrian hamster monoclonal, Biolegend).

**Bulk RNA-sequencing**. Bulk RNA-sequencing of the isolated PDGFRβ$^+$ IntSCs was performed by obtaining the alignment file. Briefly, reads were mapped using TopHat software tool. The alignment file was used to assemble transcripts, estimate their abundances, and detect differential expression of genes or isoforms using cufflinks. The Ingenuity Pathway Analysis tool (QIAGEN) was used to further evaluate the data in the context of canonical signaling and determine whether YAP/TAZ-hyperactivated genes were specifically related to the secretory molecules. The significance of the canonical signaling was tested by the Benjamini–Hochberg procedure, which adjusts the P value to correct for multiple comparisons, and their activation or inhibition was determined with reference to activation z-scores. Cluster analysis and heatmaps were generated using Morpheus (https://software.broadinstitute.org/morpheus/). For GSEA, gene set collections from the Molecular Signatures Database 4.0 (http://www.broadinstitute.org/gsea/msigdb/) were used.

**MEFs and HDLECs**. Mouse embryonic fibroblasts (MEFs) were isolated from E12.5 embryos as previously described[32]. Briefly, head, limb and heart tissues were removed and samples were minced with scissors and digested with 0.1% trypsin/EDTA with DNase (1 U/ml, Invitrogen) at 37 °C for 20 min. After digestion and removal of undigested tissues, the cells were spun briefly, plated onto a 10 cm dish, and allowed to grow to subconfluence. MEFs were then maintained in DMEM/F12 containing 10% FBS. Human dermal lymphatic endothelial cells (HDLECs) were

purchased from Lonza and cultured in endothelial growth medium (EGM2-MV, Lonza). MEFs and HDLECs were used at passage 2 or 3. Cells were incubated in a humidified atmosphere of 5% $CO_2$ at 37 °C and confirmed to be mycoplasma-negative (MycoAlert Detection Kit, Lonza).

**Spheroid-based sprouting assay**. LEC spheroids were generated by culturing HDLECs at passage 2 or 3 in culture medium containing 0.25% methylcellulose and incubating overnight as hanging drops[36]. The spheroids were then collected and embedded in 2 mg/ml collagen type I (Corning), treated with control bovine serum albumin (BSA, Sigma-Aldrich), WISP2 (300 ng/ml, Peprotech), TNFSF15 (50 ng/ml, Peprotech), BDNF (50 ng/ml, Peprotech), EBI3 (100 ng/ml, Peprotech) or ANGPT2 (5 μg/ml, in-house generated) with or without VEGF-C (200 ng/ml, R&D Systems) in Endothelial Cell Basal Medium (PromoCell) containing 1% FBS for 24 h. At the end of the incubation, spheroid was stained with cell tracker (Molecular Probes, 1.5 μM, 37 °C, 30 min). The spheroids were then fixed in 4% PFA for 15 min at RT and imaged using Cell observer (Carl Zeiss).

**Quantitative RT-PCR**. RNA was extracted using RNeasy Micro kit (Qiagen) or Trizol RNA extraction kit (Invitrogen). A total of 1 μg of extracted RNA was tran-scribed into cDNA using GoScript Reverse Transcription Kit (Promega). Quantitative real-time PCR was performed using FastStart SYBR Green Master mix (Roche) and S1000 Thermocycler (Bio-Rad) with the indicated primers. The primers were designed using Primer-BLAST or adopted from previously published studies, which are described in Supplementary Table 1. Gapdh was used as a reference gene and the results were presented as relative expressions to control. Primer reaction specificity was confirmed by melting curve analysis. Relative gene expression was analyzed by ΔΔCt method using the CFX Manager software (Bio-Rad).

**In vitro stretch and osmotic experiments**. MEFs and primary mouse IntSCs were stretched using a mechanical cell stretching instrument (STREX) with a 4% of linear stretch at 10 cycles/min, and osmotic stress was applied with 0.4 M sorbitol for 3 h as previously described[40,50]. Cells were kept in a humidified 5% $CO_2$ incubator at 37 °C during all experiments.

**Ex vivo osmotic experiments**. After opening the abdominal cavity under anes-thesia, jejunum part of small intestine was cut into 2 cm pieces. Gut fragments were opened longitudinally and washed with PBS. Tissues were then cultured in DMEM/F12 including 5% FBS and osmotic stress was applied immediately with 0.4 M sorbitol in the culture medium for 3 h. Tissues were kept in a humidified 5% CO2 incubator at 37 °C during all experiments.

**Immunofluorescence staining of MEFs**. MEFs were plated on 8-well Nunc Lab-Tek II chamber slides (Sigma-Aldrich) and fixed with 4% paraformaldehyde for 15 min at 4 °C. After several washes with PBS, samples were blocked with 5% goat (or donkey) serum in 0.5% PBST for 30 min at RT. Cells were incubated with the indicated primary antibodies at 4 °C overnight. Bound primary antibodies were detected by incubating with secondary antibodies for 90 min at RT. The following primary antibodies were used for the cell staining: anti-YAP (rabbit monoclonal, 14074, Cell signaling); anti-TAZ (rabbit polyclonal, HPA007415, Sigma-Aldrich); and Alexa Fluor 488-conjugated anti-phalloidin (A12379, Thermo Fisher) anti-bodies. Alexa Fluor 594-conjugated secondary antibodies were purchased from Jackson ImmunoResearch. Nuclei were stained with DAPI (Invitrogen) and slides were mounted and imaged as described above.

**ChIP-qPCR**. MEFs were fixed with 1% PFA for 10 min and quenched with glycine. Samples were washed with PBS and lysed with lysis buffer containing 1% SDS. The fixed DNA in the cell lysates was sonicated using Focused-ultrasonicator (Covaris). The cell lysates were centrifuged, and all but 5% (saved for whole cell lysate input control) of the resulting supernatants were diluted with ChIP dilution buffer containing 0.5% Triton X-100. Diluted samples were then incubated overnight at 4 °C with anti-TEAD4 antibody (mouse monoclonal, ab58310, Abcam). Then, protein A/G Dynabeads (Thermo Fisher) were added and the samples were incubated for 2 h at 4 °C. The beads were isolated with DynaMag-2 (Thermo Fisher), washed with series of ChIP wash buffers: low-salt wash buffer, high-salt wash buffer, LiCl wash buffer, and TE buffer. Samples were eluted in 1% SDS two times for 15 min each at 65 °C. The beads were removed and both the eluted material and 5% whole cell lysate input samples were reverse-cross-linked over-night at 65 °C. After normalizing the pH, samples were incubated for 30 min with RNase (3 μg/mL) at 37 °C and for 2 h with proteinase K (20 mg/ml) and glycogen (20 mg/ml) at 55 °C. DNA was eluted using standard procedures and ChIP-DNA were analyzed by qPCR compared with input samples using the primers listed in Supplementary Table 2.

**ELISA**. Supernatant of primary cultured mouse IntSCs were harvested after stretching, and supernatant was centrifugation for 15 min. The concentration of VEGF-C in supernatant was measured by enzyme-linked immunosorbent assay (ELISA) with mouse VEGF-C–specific ELISA kit (CSB-E07361m, Cusabio).

**Morphometric analysis**. Morphometric measurements were performed using ImageJ software (NIH), ZEN 2012 software (Carl Zeiss), or Imaris (Bitplane). For lacteal surface area quantification, a threshold was set for the images to be analyzed and absolute villus LYVE-1$^+$ area was measured for each lacteal within the villus. Absolute lacteal length, absolute villi length, and absolute villi width were measured using images of villus E-cadherin$^+$ or CD31$^+$ or LYVE-1$^+$ immunostaining in each random 850 μm$^2$ fields of the indicated part of the intestine. For lacteal and villi surface area, length, and width quantifications, at least 100 villi were analyzed along the whole small intestine. Quantification for the following parameters in the small intestine were performed in jejunum. Number of lacteal sprouts and number of lacteal branching were measured in 10-20 random LYVE-1$^+$ lacteal. Number of Prox1$^+$ LECs and number of Prox1$^+$EdU$^+$ LECs were counted manually within 100 μm length of 10–20 random LYVE-1$^+$ lacteal. Quantification of VE-cadherin$^+$ junctions of lacteal was performed with ×40 objective lens in lacteals of 5–10 villi per mouse. Briefly, zipper-type junction was defined as continuous junctions at cell–cell borders with cells that have elongated shape[51]. Button-like junction was defined as discontinuous junctions that are not parallel with the cell–cell borders and the oak leaf-like cell shape. Junctions that do not match either pattern were categorized as mixed type. Fluorescence intensity (FI) of BODIPY and its quantification was performed as previously described[34]. Oil red O area was measured as Oil red O$^+$ area divided by villus area and normalized by the average of those of control mice. Villus smooth muscle cell coverage area was measured as absolute αSMA$^+$ coverage area in five 850 μm$^2$ fields per sample, which was then normalized by the average of those of control mice. To quantify the relative expressions of lacteal VEGFR3, mean FI measured in five 425 μm$^2$ fields per samples and was presented as relative values divided by its control. To determine blood vessel density, area of VEGFR2 was measured in five 425 μm$^2$ fields per sample and presented as relative values to its control. The area of CD3$^+$ T cell or F4/80$^+$ macrophage was measured in five 425 μm$^2$ fields per samples. Lymphatic vessel density was calculated by measuring LYVE-1$^+$ area in five random 425–850 μm$^2$ fields from whole mount samples and presented as relative values to its control.

**Droplet-based single-cell RNA sequencing**. Sorted single cells were processed using the 10X Chromium Single cell 3′ Reagent Kit v3 (10X genomics) following manufacturer's protocols. Briefly, sorted cells were resuspended in PBS with 0.5% BSA and mixed with RT reagent mix and RT primer, which were then added to each channel in 10X chips, targeting 5000 cells. Cells were separated into Gel Beads in Emulsion where RNA transcripts from single cells were barcoded and reverse-transcribed. After cDNA library construction and amplification, cDNA molecules were enzymatically fragmented, end-repaired and A-tailed. After selecting the appropriate size of processed cDNA molecules through double-sized size selection using SPRI beads (Beckman Coulter), they were ligated with an adaptor, and sample-index PCR was performed. After another double-sized size selection using SPRI beads, final library constructs were diluted 10-fold and ran on the Agilent Bioanalyzer High Sensitivity Chip for quality control. Single-cell libraries were then sequenced using Illumina HiSeq-X platform.

**Pre-processing of single-cell RNA data**. Raw sequencing data were first de-multiplexed and mapped to mouse reference genome (mm10) by using Cell Ranger 3.0.2 toolkit from 10X Genomics (http://10xgenomics.com). Using R package Seurat (version 3.0.0)[52], raw expression matrices were built using Read10X function. Cells with fewer than 1000 detected genes or higher than 6000 detected genes (considered as potential doublets) were excluded (Supplementary Fig. 17a). In addition, cells with high mitochondrial gene associated unique molecular identifier (UMI) counts (>7% of total) were considered as dead cells and thus excluded. For genes, those expressed by fewer than 3 cells were removed. In addition, a small number of contaminating Ptprc$^+$ immune cells and Pecam1$^+$ Cdh5$^+$ endothelial cells were excluded after initial round of clustering. After removing unwanted cells and genes, UMI counts for each gene for a cell were divided by the total expression of a given cell and multiplied by 10,000 and log-transformed. Then, genes were scaled and centered by regressing out variables such as mitochondrial gene percentage and number of UMIs.

**Identifying variable genes and dimensionality reduction**. R package Seurat was used for clustering analysis. First, highly variable genes across dataset were identified using FindVariableFeatures function in Seurat with option: selection.method = "vst". Top 2000 genes with highest variability were selected for downstream analysis. Using the identified variable genes, initial dimensionality reductions were performed using principal component analysis (PCA). JackStraw function was used to determine statistical significance of PCA scores. The most significant 30 principal components (PCs) were used as input for Uniform Manifold Approximation and Projection (UMAP) to reduce the data into two-dimensional space.

**Cluster analysis**. In order to cluster cells by their transcriptional profiles, we performed an unsupervised clustering on top 2000 highly variable genes for all four samples. We first built shared nearest neighbors (SNN) graph on the principal component space by using FindNeighbors function. Then applied louvain algorithm for modularity optimization-based clustering by using FindClusters function. Clustering resolution parameters were set at the point where clusters showed highly

distinct transcriptional profiles. Clusterings were independent to the number of genes detected per cell (Supplementary Fig. 17b). Cluster-specific markers were differentially expressed genes for each cluster identified through FindMarkers function in Seurat with the following settings: min.pct = 0.3, logfc.threshold = 0.25, min.diff.pct = 0.15, test.use = "MAST". Identified markers with adjusted $P < 0.05$ were used for further analysis. Since some cell types such as smooth muscle cells and mural cells had well-known markers, their appearance at the top of the marker list for each cluster validated our marker selection processes. To remove unwanted source of variations such as sex, cells were split by presence female specific transcript $Xist$ and integrated. To perform integration of different datasets, we first log normalized each raw expression matrices and identified top 2000 highly variable genes for each dataset. Then, using FindIntegrationAnchors function in Seurat, we identified anchors between datasets. Next, datasets were integrated based on the identified anchors via IntegrateData function in Seurat. Integrated datasets were then scaled and dimensions were reduced for further cluster analysis. Clusters in the integrated datasets were annotated by their expression profiles of genes in the identified maker lists.

**Gene ontology enrichment analysis**. Gene ontology enrichment analysis on the cluster markers were performed using R package goseq (version 1.34.0)[53]. Depending on the gene's length, weightings for each gene were obtained and the Wallenius approximation was used to identify associated gene ontology terms. Gene ontology terms that have adjusted $P < 0.05$ and biological process categories were selected.

**Statistics**. No statistical methods were used to predetermine sample size. The experiments were randomized and investigators were blinded to allocation during experiments and outcome analyses. All values are presented as mean ± standard deviation (SD). Statistical significance was determined by the two-sided Mann–Whitney $U$ test between two groups or the two-way ANOVA with Holm–Sidak's multiple comparisons test for multiple-group comparison. Statistical analyses were performed using GraphPad Prism 8.0 (GraphPad Software). Statistical significance was set at $P < 0.05$.

**Reporting summary**. Further information on research design is available in the Nature Research Reporting Summary linked to this article.

## Data availability

The source data underlying all main and Supplementary figures are provided as a Source Data file. RNA sequencing data including bulk and single-cell data are available in the National Center for Biotechnology Information's Gene Expression Omnibus under accession number GSE124488. All the other materials are available from the corresponding author upon reasonable request.

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

## Acknowledgements
This study was supported by the Institute for Basic Science funded by the Ministry of Science, ICT and Future Planning, Korea (IBS-R025-D1-2015 to G.Y.K.). We thank Pilhan Kim and Young Seok Ju (KAIST) for helping intravital imaging and bioinformatical analysis; Sungyong Park, Sujin Seo, Jeomil Bae, Duri Choi and Hyun-Tae Kim (IBS) for technical assistances.

## Author contributions
S.P.H. and G.Y.K. conceived and designed the study. S.P.H. performed all mouse work and most of the experiments with contributions from M.J.Y., H.C., H.B., K.C., S.H.S., and I.P. M.J.Y. performed single-cell RNA sequencing, and analyzed single-cell RNA-seq datasets. H.C. performed in vitro ChIP-qPCR experiment. K.C. performed intravital imaging. H.B. analyzed bulk RNA-seq datasets. S.H.S. helped with mouse work and in vivo data analysis. K.A., R.H.A., and D.L. provided mice and reagents, and reviewed the manuscript. H.C. wrote the manuscript with contributions from S.P.H., M.J.Y., I.P. under the supervision of G.Y.K. I.P. proofread and edited the manuscript. S.P.H., M.J.Y., H.C., and G.Y.K. analyzed and interpreted data, and generated figures. S.P.H., M.J.Y., H. C. actively discussed the experiments and the results under the supervision of G.Y.K. All authors discussed the results and commented on the manuscript. G.Y.K. supervised and directed the project.

## Competing interests
The authors declare no competing interests.
