## [Peer Review File · Nature Communications]

Reviewers' comments:

Reviewer #1 (Remarks to the Author):

This is an exceptionally well-executed and well-assembled study exploring a very important question in the field of lymphatic biology: What is the source of VEGFC for lymphangiogenesis. The authors have used sophisticated genetic models to argue that a subset of PDGFR β + ISCs within the intestinal lacteal are required for secreting VEGFC, in response to mechanical or osmotic stress, in order to maintain lacteal integrity. Single-cell RNA Seq further characterizes the heterogeneity of villus ISCs--which is a data set that will be valuable to the broader field of intestinal biology. The data are mostly convincing and the findings are certainly of interest. Statistics are appropriate and correct. However, addressing the following points would help to strengthen the claims and overall significance of the paper:

1. A limitation to the study is the artificial nature of the Lats1/2 hyper-activation mutants, particularly since Yap/Taz are only modestly expressed in the wildtype PDGFR+ ISC. I don't think there is much that the authors can do about this, other than to clarify and explain for readers the limitations of potential over-interpretation that can come from a transgenic over-expression model.
2. A prior report by this group also demonstrated an essential role for immune cells, particularly macrophages, within the lacteal/villus niche, as a key source of VEGFC required for lacteal integrity. EMBO Rep (2019)20:e46927. This leaves one to question which cell types—macrophages or fibroblasts—are relatively most important. Extended Figure 2d shows immunohistochemical staining for lymphatics that appears to show a fairly significant reduction in macrophage number (at least visually, based on the images provided). The authors are encouraged to provide more rigorous quantitation of macrophage numbers, as they performed by FACS analysis in EMBO Rep (2019)20:e46927.
3. Along these lines, if indeed the original conclusion of "no change in macrophage number" is observed, then why don't the macrophages compensate for the reduction in VEGFC in the Yap/Tazi Δ P β C mice (Fig. 4 and extended data Fig 7d)?
4. Results of the ip AAV VEGFR3 blockade in Fig. 3 are expected and consistent with a prior publication from this group EMBO Rep (2019)20:e46927 reporting LEC-specific VEGFR3 gene deletion. So, it is unclear how this data is conceptually advancing the current study. Perhaps better explanation, or removal of this data should be considered.
5. A key conceptual and mechanistic paradigm put forth by some of the in vitro studies in this paper is that "osmotic or mechanical stretch" regulated the YAP/TAZ-pathway stimulates VEGFC secretion. However, this is not demonstrated using in vivo models. What evidence do the authors provide to validate that mechanical stretch during normal or abnormal physiological processes is actually engaged, at a molecular level, in lacteals/ISCs? The impact and significance of the study would be greatly improved if "mechanical or osmotic stress" were experimentally induced in the genetic animals.
6. Perhaps I missed this in the supplemental materials, but what is the phenotype of Yap/Tazi Δ P β C mice with regard to weight gain/ intestinal function? What happens to these animals under conditions of intestinal/lacteal challenge?
7. Spatial localization of selected genes is not particularly convincing in Figure 6h. The authors are encouraged to split the channels, or use other approaches, to improve the quality and interpretation of this data.

Reviewer #2 (Remarks to the Author):

This manuscript addresses an important issue in lymphatic vascular biology relating to the function and remodelling of the lacteal vessels in the intestine. The study employs state-of-the-art genetic mouse models to compellingly demonstrate the role of fibroblast subsets, the YAP/TAZ signalling system and VEGF-C/VEGFR-3 paracrine signalling in the structural and functional modulation of these important lymphatic vessels. The data are novel, well presented and extensive, and support all conclusions of the study. The statistical approaches appear appropriate, and sufficient detail is provided to allow the study to be repeated. Overall, the study is creative, thorough and elegant - I could not find fault with it. Unusually, I have no suggestions for improvement. The manuscript would be of major interest to, and shape the thinking of, lymphatic vascular biologists and researchers/clinicians focussed on intestinal biology more broadly.

Reviewer #3 (Remarks to the Author):

The authors present studies exploring the role of the Hippo pathway in regulating the function of intestinal stromal cells and the lacteal system in intestinal villi. They employ *Pdgfrb*-directed CreERT2 to deleted *Lats1/2* and *Yap/Taz* and assess alterations in the lacteal system. In particular *Lats1/2* deletion leads to increased lacteal sprouting and branching that causes altered cell-cell junctional organization, and defective fat absorption. The authors propose that this is through Hippo-dependent regulation in a novel subset of *Pdgfrb*⁺ stromal cells. Overall, this is a very interesting manuscript, the quality of the data is very high, and I support publication pending some specific, relatively minor revisions:

1) Much is made of the *Pdgfrb*-Cre driving deletion in pericytes. Most of the experiments use TdTomato lineage tracing. Confirming that TdTom expression accurately marks endogenous *Pdgfrb*-expressing pericyte populations, for example by counterstaining using smFISH, is important.

2) To confirm that the *Lats1/2* deletion alters lacteal morphogenesis via *Yap/Taz*, the authors need to test if concomitant loss of YAP/TAZ suppresses the phenotype.

3). Similarly, the authors show that *Vegfc* is induced by *Lats* DKO and then go on to show that interference with *Vegfc*:*Vegfr3* signaling using a AAV-mVEGFR34-7-Ig strategy to express the extracellular domain of *Vegfr3*, suppresses the lacteal phenotype. This is a very elegant experiment that certainly supports the model. It would be ideal if the authors also tested if knock out of *Vegfc* floxed allele in *Pdgfrb*-Cre expressing cells suppresses the *Lats* DKO phenotype.

4) The authors go on to nicely demonstrate regulation of this pathway by mechanotransduction mechanisms and conclude with single cell analysis of *Pdgfrb* lineage-traced cells to try and identify the key fibroblast subtype that mediates *Vegfc* expression and lacteal morphogenesis. They identify novel subsets that may be the key targets, but this line of investigation ends abruptly, and the analysis is a bit superficial in my opinion. In its current form I am not sure it adds a huge amount to the manuscript as it opens up more questions and there are many more analyses required. A number of questions need addressing:

-it would be useful to know how many stromal cells are *Pdgfrb*-positive. Comparison to a recently published single cell profile of intestinal mesenchyme would be useful here (PMID: 31953387).

-the authors identify a vFB1 population that are very similar to vFB2, except that vFB1 are marked by *Fos* and *Jun*. vFB1 may reflect a phenotypic artefact of cell stress induced by dissociation. Analysis of stress genes should be included and the existence of this cell type clarified by testing if vFB1 exist in vivo.

-the authors show some in situ studies but the quality is poor and the key question; are they

distinct cell types? needs co-staining with multiple markers and imaging at single cell resolution.

- the authors should validate in vivo by in situ or IHC that Vegfc is regulated by Lats1/2 and Yap/Taz in the novel vFB populations to distinguish if all three vFBs regulate Vegfc
- vFB1-3 don't seem very distinct in the UMAP. What do the distinct vFB1-3 signature gene profiles look like across all cells? My concern is whether these are really distinct cell types as proposed by the authors. For example, Vegfc looks like it marks most of vFB3, but only subsets of vFB1-2 populations. More detailed clustering of the vFB1-4 populations would be helpful in defining what's going on with these populations.
- a subset of FB4 (Grem+;Ackr4+) are also marked by Vegfc, so FB4 might contribute to lacteal integrity, not just FB1-3 as proposed by the authors. This needs to be examined more carefully.
- what's the cell type abundance in WT versus Lats DKO?

Reviewer #4 (Remarks to the Author):

General comments:

The authors have conducted a well-conceived and thorough study into the role of YAP/TAZ signaling in intestinal stromal cells. This is a very nice study with a substantial amount of evidence supporting the role of villi fibroblasts in relation to lacteal integrity.

Of note, the application of scRNA-seq to examine cell heterogeneity allowed the detection of several important subtypes with apparently different functional roles. This information was used to define the function and location of several different, and novel, cell types. This detail is a major advancement that will promote in-depth studies of these intestinal cells in both health and disease.

Overall, this was an enjoyable paper to read and I congratulate the authors on their efforts.

Despite the massive amount of work, I can't help but feel that the scRNA-seq was merely an afterthought and there seems to be little attempt to validate any of the novel subpopulations using orthogonal methods. My main constructive criticism here is that it is crucial to validate any expression data, particularly scRNA-seq, but not much was done to validate the cell types identified here. There are multiple robust wet-lab methods available to define the expression of population-specific marker genes at the single cell level - smRNA-FISH being a popular alternative to standard IHC.

Another suggestion is to consider silencing or deleting these subpopulations, either as a cell line or in vivo, to better understand their overall effect.

Detailed comments:

Authors need to please check the methods for the genomics components of the study. As far as this reviewer is aware, the Illumina Hi-Seq-X is not compatible with RNA-seq libraries of any kind - it is more likely that a different instrument was used, such as a HiSeq 4000 or NovaSeq. Following this, please add more details about the sequencing run configuration, sequencing read depth per library and the subsequent read depth per cell. This detail is important to understand the quality of the dataset being presented.

I noted the lower threshold of 'detected genes per cell' as 1000, which is actually quite high. How was this value arrived at? In typical situations, 200-500 is sufficient. I'm curious to know whether the extra 500 genes per cell would influence the cluster cell-type classifications. Have you tried to alter these thresholds and look at the outcomes here?

Regarding the differential expression tests, how did the authors settle on using MAST as the DE test method? Were other methods also used? If so, how did the results appear?

It would be fascinating to use a knockout mouse for any one of those key marker genes to see if there is an obvious phenotypic effect on the lacteal. The model(s) could then be used for additional scRNA-seq comparisons and/or methods such as single molecule RNA-FISH to more clearly show how these populations are functioning.

One important avenue that was not explored in this work was that of the human translation. Are there any human single cell datasets or tissue samples available that any of these novel populations can also be identified in? I believe this to be probably one of the most important aspects of this study; should these populations be identified and functionally understood in humans, I expect there would be huge implications for GI health.

Finally, please make the datasets available to the public - I note that you have not yet uploaded the data to a repository, or made it available to the reviewers.

Detailed Point-by-Point Response to Reviewers' Comments

We deeply appreciate the editor and reviewers for their thoughtful, critical, and constructive comments, which have undoubtedly provided us with valuable opportunities to improve our work. We have performed additional experiments and revised the manuscript to address the issues raised by the reviewers. To readily keep track of the changes that were made in revising our manuscript, we have highlighted them in blue except for the changes of figure legends.

Referee #1

This is an exceptionally well-executed and well-assembled study exploring a very important question in the field of lymphatic biology: What is the source of VEGFC for lymphangiogenesis. The authors have used sophisticated genetic models to argue that a subset of PDGFR β + ISCs within the intestinal lacteal are required for secreting VEGFC, in response to mechanical or osmotic stress, in order to maintain lacteal integrity. Single-cell RNA Seq further characterizes the heterogeneity of villus ISCs--which is a data set that will be valuable to the broader field of intestinal biology. The data are mostly convincing and the findings are certainly of interest. Statistics are appropriate and correct. However, addressing the following points would help to strengthen the claims and overall significance of the paper:

We appreciate these favorable and encouraging comments.

Comment 1: A limitation to the study is the artificial nature of the Lats1/2 hyper-activation mutants, particularly since Yap/Taz are only modestly expressed in the wild type PDGFR+ ISC. I don't think there is much that the authors can do about this, other than to clarify and explain for readers the limitations of potential over-interpretation that can come from a transgenic over-expression model.

Response: We appreciate this concern. Although we have observed abnormal lacteal sprouting and branching through YAP/TAZ activation in intestinal fibroblasts, limitations exist due to the artificial nature in utilizing genetically modified mouse models. Thus, we admit the limitation of this study until it represents a pathophysiological condition in which YAP/TAZ is activated in intestinal fibroblasts. We have explained about the limitations of this study into the revised manuscript accordingly as below.

Discussion (Page 21) However, coupling our phenotypic observations to pathophysiological conditions needs further investigation since we used mouse models in which YAP/TAZ are artificially activated.

Comment 2: A prior report by this group also demonstrated an essential role for immune cells, particularly macrophages, within the lacteal/villus niche, as a key source of VEGFC required for lacteal integrity. EMBO Rep (2019)20:e46927. This leaves one to question which cell types—macrophages or fibroblasts—are relatively most important. Extended Figure 2d shows immunohistochemical staining for lymphatics that appears to show a fairly significant reduction in macrophage number (at least visually, based on the images provided). The authors are encouraged to provide more rigorous quantitation of macrophage numbers, as they performed by FACs analysis in EMBO Rep (2019)20:e46927.

Response: We appreciate this comment. We acknowledge that macrophages are also a source of VEGF-C that maintain lacteal integrity. Therefore, to evaluate the changes in macrophages, we performed flow cytometric analysis to quantify the macrophage population. Flow cytometric analysis indicated no changes in MHCII⁺ F4/80⁺ macrophage numbers between WT and *Lats1/2*^{iΔPβC} mice (For Reviewer Only, Fig. 1). Thus, we believe that YAP/TAZ-mediated VEGF-C secretion by intestinal fibroblasts is the main regulator of lacteal integrity and is independent of—and relatively more important than—macrophages.

For Reviewer Only, Fig. 1 Flow cytometric analysis of MHCII⁺ F4/80⁺ macrophages in WT and *Lats1/2*^{iΔPβC} mice. **a,b**, Representative flow cytometric analysis and comparison of MHCII⁺ F4/80⁺ macrophage gated on DAPI⁻ CD45⁺ cells from small intestine of WT and *Lats1/2*^{iΔPβC} mice. Each dot indicates a value from *n* = 4 mice/group. Horizontal bars indicate mean ± SD and *P* value versus WT by Mann-Whitney U test. n.s., not significant.

Comment 3: Along these lines, if indeed the original conclusion of “no change in macrophage number” is observed, then why don’t the macrophages compensate for the reduction in VEGFC in the *Yap/Tazi*^{ΔPβC} mice (Fig. 4 and extended data Fig 7d)?

Response: Please note that this response also addresses Comment 2. We speculate that this is due to a different regulatory mechanism of VEGF-C secretion between fibroblasts and macrophages. In case of PDGFRβ⁺ fibroblasts, our results indicate that mechanical and osmotic stress regulate YAP/TAZ expression to maintain VEGF-C secretion. On the other hand, macrophages are reliant on intestinal microbiota to

secrete VEGF-C (Suh et al., *EMBO Rep.*, 2019). Moreover, fibroblasts and macrophages do not have any known sort of a compensatory 'interactive feedback' function that detects VEGF-C level or subsequent downstream, and therefore presumably lacks the basic feature of a compensatory mechanism. Thus, we believe that the macrophages or the fibroblasts cannot compensate for changes in VEGF-C secretion due to the differences in the regulatory mechanism for VEGF-C secretion.

Comment 4: Results of the ip AAV VEGFR3 blockade in Fig. 3 are expected and consistent with a prior publication from this group EMBO Rep (2019)20:e46927 reporting LEC-specific VEGFR3 gene deletion. So, it is unclear how this data is conceptually advancing the current study. Perhaps better explanation, or removal of this data should be considered.

Response: Our intention of the AAV-VEGFR3 blockade study was to investigate whether VEGF-C secretion by PDGFR β ⁺ ISCs is responsible for the aberrant lacteal phenotype. Through the AAV-VEGFR3 blockade experiments, we could demonstrate that VEGFR3 activation, but not other pathways, is the main factor that induces aberrant sprouting and branching of lacteals, as deductible by the complete restoration of lacteal structure. This was a critical and logical step that directed us to investigate whether YAP/TAZ activation in PDGFR β ⁺ ISCs induces VEGF-C secretion (Figure 5). Therefore, we believe that this result is important to help readers understand the reasoning behind the design of subsequent experiments that identify VEGF-C as the main secretory molecule that is induced by YAP/TAZ activation and is responsible for lacteal branching and sprouting in our model.

Comment 5: A key conceptual and mechanistic paradigm put forth by some of the in vitro studies in this paper is that "osmotic or mechanical stretch" regulated the YAP/TAZ-pathway stimulates VEGFC secretion. However, this is not demonstrated using in vivo models. What evidence do the authors provide to validate that mechanical stretch during normal or abnormal physiological processes is actually engaged, at a molecular level, in lacteals/ISCs? The impact and significance of the study would be greatly improved if "mechanical or osmotic stress" were experimentally induced in the genetic animals.

Response: We appreciate this constructive suggestion. To address this point, we investigated the effect of osmotic stress *in vivo* by adapting a reported method (Tropini *et al.*, *Cell*, 2018). Osmotic stress not only reduced the levels of YAP and TAZ but also changed their localization to the cytoplasm compared with control in the PDGFR β ⁺ ISCs (For Reviewer Only, Fig. 2a,b). On the other hand, there is no suitable *in vivo* method for applying mechanical stress onto the small intestine at this moment. Although we employed a stretching instrument (STREX) to induce mechanical stress onto the small intestine, it did not seem to induce mechanical stress on the intestinal villi. Therefore, we would like develop these techniques and

study the consequences in a future study.

For Reviewer Only, Fig. 2 Osmotic stress reduces the levels of YAP/TAZ and changes their localization to the cytoplasm in $PDGFR\beta^+$ ISCs *in vivo*. **a**, A schematic diagram depicting the induction of osmotic stress in small intestine of WT mice by administration of 20% PEG dissolved in drinking water in 8-week-old mice for 7 days. **b**, Representative images and comparisons of YAP/TAZ levels and their localizations in $PDGFR\beta^+$ ISCs of control or osmotic stress-applied villi; insets show magnified view. Similar findings were observed in four independent experiments using $n = 4$ mice. Scale bars, 25 μm .

Comment 6: Perhaps I missed this is the supplemental materials, but what is the phenotype of $Yap/Tazi\Delta P\beta C$ mice with regard to weight gain/ intestinal function? What happens to these animals under conditional of intestinal/lacteal challenge?

Response: We appreciate this comment. In Fig. 4f, we showed that lipid absorption

of lacteals is impaired in *Yap/Taz*^{iΔPβC} mice by measuring plasma triglycerides after olive oil gavage. In addition to this result, we additionally challenged these mice with high-fat diet (HFD) for 8 weeks in order to observe weight gain/intestinal function. After HFD feeding, *Yap/Taz*^{iΔPβC} mice gained significantly less weight, exhibited less fat mass and less plasma triglycerides, and had less ectopic fat accumulation in the liver compared with WT mice (Extended Data Fig. 10b-g). In contrast, we observed no difference in weight gain between WT and *Yap/Taz*^{iΔPβC} mice during normal chow diet (Extended Data Fig. 10a). These findings indicate that *Yap/Taz*^{iΔPβC} mice have impaired lipid absorption, presumably due to lacteal dysfunction. We included these additional results and their descriptions into the revised manuscript accordingly as below.

Results (Page 12) We also investigated whether lacteal dysfunction in *Yap/Taz*^{iΔPβC} mice has an effect on dietary lipid absorption in *Yap/Taz*^{iΔPβC} mice with high-fat diet (HFD) challenge for 8 weeks. While we observed no difference in weight gain during normal chow (NC) diet, *Yap/Taz*^{iΔPβC} mice gained significantly less weight, exhibited less fat mass and less plasma triglycerides, and had less ectopic fat accumulation in the liver compared with WT mice (Extended Data Fig. 10b-g). These findings suggest that *Yap/Taz*^{iΔPβC} mice have impaired lipid absorption, presumably due to lacteal dysfunction.

Extended Data Fig. 10 Impaired dietary lipid absorption in *Yap/Taz*^{iAPβC} mice. **a**, Comparison of weight gain of normal chow (NC)-fed WT and *Yap/Taz*^{iAPβC} mice. Each dot indicates a value from $n = 25$ mice/group. Horizontal bars indicate mean \pm SD and P value versus WT by Mann-Whitney U test. n.s., not significant. **b**, Gross morphology of WT and *Yap/Taz*^{iAPβC} mice after 8 weeks of high-fat diet (HFD). Scale bars, 1 cm. **c**, Representative gross image of inguinal white adipose tissue (WAT) and gonadal WAT of WT and *Yap/Taz*^{iAPβC} mice after 8 weeks of HFD. Scale bars, 1 cm. **d**, Comparison of body weight of WT and *Yap/Taz*^{iAPβC} mice after 8 weeks of HFD. Each dot indicates a value from $n = 8$ to 15 mice/group. Horizontal bars indicate mean \pm SD and P value versus WT by Mann-Whitney U test. **e**, Comparison of plasma triglyceride concentration (mg/dL) in 6 hr fasted WT and *Yap/Taz*^{iAPβC} mice after 8 weeks of HFD. Each dot indicates a value from $n = 8$ mice/group. Horizontal bars indicate mean \pm SD and P value versus WT by Mann-Whitney U test. **f**, **g**, Representative image and comparison of Oil red O staining in liver of WT and *Yap/Taz*^{iAPβC} mice after 8 weeks of HFD. Each dot indicates a mean value taken from five different regions of a mouse and $n = 5$ mice/group. Horizontal bars indicate mean \pm SD and P value versus WT by Mann-Whitney U test. Scale bars, 50 μ m.

Comment 7: Spatial localization of selected genes is not particularly convincing in Figure 6h. The authors are encouraged to split the channels, or use other approaches, to improve the quality and interpretation of this data.

Response: Thank you for this constructive comment. Please note that this issue was also discussed in Reviewer 3's Comment 4-3 and Reviewer 4's General comments. To reveal the identity of novel subpopulations that we found, we performed additional immunofluorescence co-staining of the distinctively expressed marker genes identified from scRNA-Seq. Although vFB1, 2, and 3 shared the expression of several genes such as *Serpina3N*, we were able to distinguish vFB3 from vFB1 and 2 by the specific expression of *P2X1* in vFB3 (Fig. 7a). In addition, while vFB1 and 2 commonly expressed *Shisa3*, vFB1 distinctly *Fosb* (Fig. 7b). Through additional immunofluorescence studies, we were able to validate existence and distinctive nature of fibroblast heterogeneity composing the small intestine lamina propria (LP). We included these additional results and their descriptions into the revised manuscript accordingly as below.

Results (Page 18) To define the location of these newly identified subsets of fibroblasts, we examined and validated the expressions of markers that are specific to each cluster (Fig. 6h,i). Of note, vFB1-3 were distributed in close proximity to the lacteal, while *Ackr4*⁺ FB4 and *Sox6*⁺ FB5 were localized around the submucosal and subepithelial regions, respectively (Fig. 6h,i), as previously reported^{12, 41}. **vFB1-3 were distinguished by specific markers: *P2X1* was exclusively expressed by vFB3 but not by vFB1-2; *Fosb* was specifically expressed by vFB1 but not by vFB2 (Fig. 7a,b).**

Figure 7. vFB1-3 are distinct cell types secreting VEGF-C. a, Representative images of vFB3 ($P2X1^+Serpin3n^+PDGFR\beta^+$) and vFB1-2 ($P2X1^-Serpin3n^+PDGFR\beta^+$) in WT mouse. White and yellow dotted boxes are magnified in the upper right and lower right panels, respectively. Red and yellow arrowheads indicate vFB3 and vFB1-2, respectively. Scale bar, 20 μ m. **b,** Representative images of vFB1 ($Serpin3n^+Fosb^+Shisa3^+PDGFR\beta^+$) and vFB2 ($Serpin3n^+Fosb^-Shisa3^+PDGFR\beta^+$) in WT mouse. White and yellow dotted boxes are magnified in the upper right and lower right panels, respectively, for each image. White and yellow arrowheads indicate vFB1 and vFB2, respectively. Scale bars, 20 μ m.

***References for responses to comments of Referee #1**

Suh SH et al., (2019). Gut microbiota regulates lacteal integrity by inducing VEGF-C in intestinal villus macrophages. *EMBO Rep.* Apr 20: e46927

Carolina Tropini et al., (2018). Transient Osmotic Perturbation Causes Long-Term Alteration to the Gut Microbiota. *Cell.* June 14;173(7):1742-1754.

Referee #2

This manuscript addresses an important issue in lymphatic vascular biology relating to the function and remodelling of the lacteal vessels in the intestine. The study employs state-of-the-art genetic mouse models to compellingly demonstrate the role of fibroblast subsets, the YAP/TAZ signalling system and VEGF-C/VEGFR-3 paracrine signalling in the structural and functional modulation of these important lymphatic vessels. The data are novel, well presented and extensive, and support all conclusions of the study. The statistical approaches appear appropriate, and sufficient detail is provided to allow the study to be repeated. Overall, the study is creative, thorough and elegant - I could not find fault with it. Unusually, I have no suggestions for improvement. The manuscript would be of major interest to, and shape the thinking of, lymphatic vascular biologists and researchers/clinicians focussed on intestinal biology more broadly.

We appreciate these unusual yet favorable and encouraging comments. We are also eager to share our study and excited to see how it could contribute to lymphatic research.

Referee #3

The authors present studies exploring the role of the Hippo pathway in regulating the function of intestinal stromal cells and the lacteal system in intestinal villi. They employ *Pdgfrb*-directed CreERT2 to deleted *Lats1/2* and *Yap/Taz* and assess alterations in the lacteal system. In particular *Lats1/2* deletion leads to increased lacteal sprouting and branching that causes altered cell-cell junctional organization, and defective fat absorption. The authors propose that this is through Hippo-dependent regulation in a novel subset of *Pdgfrb*⁺ stromal cells. Overall, this is a very interesting manuscript, the quality of the data is very high, and I support publication pending some specific, relatively minor revisions:

We appreciate these favorable and encouraging comments.

Comment 1: Much is made of the *Pdg* lineage tracing. Confirming that TdTom expression accurately marks endogenous *Pdgfrb*-expressing *frb*-Cre driving deletion in pericytes. Most of the experiments use TdTomato pericyte populations, for example by counterstaining using smFISH, is important.

Response: The *Pdgfrb*-cre-ER^{T2} mice have been validated in previous studies that studied the roles of pericytes in the retina and lung (Park et al., *Nat Commun.*, 2017; Kato et al., *Nat Commun.* 2018). Since ablation of *Pdgfrb*-cre population by diphtheria toxin A completely depleted PDGFRβ⁺NG2⁺ pericytes (Park et al., *Nat Commun.*, 2017), the *Pdgfrb*-cre-ER^{T2} mice accurately marks endogenous pericyte population. The deletion efficiency of other PDGFRβ⁺ populations such as fibroblasts and smooth muscle cells has also been confirmed (Choi et al., *Nat Commun.*, 2020; Park et al., *Nat Commun.*, 2017; Kato et al., *Nat Commun.* 2018). Thus, we believe that the *Pdgfrb*-cre-ER^{T2} mice we used in this study has been sufficiently validated in several studies for its labeling and deletion efficiency of PDGFRβ⁺ population including pericytes, fibroblasts, and smooth muscle cells.

Comment 2: To confirm that the *Lats1/2* deletion alters lacteal morphogenesis via *Yap/Taz*, the authors need to test if concomitant loss of YAP/TAZ suppresses the phenotype.

Response: This is valid point that need to be addressed. To confirm that LATS1/2 deletion alters lacteal morphogenesis via YAP/TAZ, we investigated the lacteal morphology in *Lats1/2-Yap/Taz*^{ΔPβC} mice (Extended Data Fig. 2a). The aberrant sprouting and branching phenotypes in lacteals of *Lats1/2*^{ΔPβC} mice were restored in *Lats1/2-Yap/Taz*^{ΔPβC} mice (Extended Data Fig. 2b). This result indicates that LATS1/2 deletion alters lacteal morphogenesis through YAP/TAZ. We included these additional results and their descriptions into the revised manuscript accordingly as below.

Results (Page 18) Moreover, to confirm that LATS1/2 deletion induces aberrant lacteal phenotypes via YAP/TAZ, we observed the lacteal phenotypes using *Lats1/2-Yap/Taz*^{iAPBC} mice (Extended Data Fig. 2a). The aberrant sprouting and branching phenotypes in lacteals of *Lats1/2*^{iAPBC} mice were restored in *Lats1/2-Yap/Taz*^{iAPBC} mice (Extended Data Fig. 2b). This result indicates that LATS1/2 deletion alters lacteal morphogenesis through YAP/TAZ.

Extended Data Fig. 2 Rescue of aberrant lacteal phenotypes in *Lats1/2-Yap/Taz*^{iAPBC} mice. **a**, Diagram depicting the generation of *Lats1/2-Yap/Taz*^{iAPBC} mice and PDGFRβ⁺ cell-specific depletion of *Lats1/2* or *Lats1/2-Yap/Taz* by tamoxifen administration in 8-week-old mice and analyses at 2 weeks later. **b**, Representative images of LYVE-1⁺ lacteals, CD31⁺ capillary plexus, and PDGFRβ⁺ stromal cells in duodenum (DD), jejunum (JJ), and ileum (IL) of small intestine in WT, *Lats1/2*^{iAPBC}, and *Lats1/2-Yap/Taz*^{iAPBC} mice. Similar findings were observed in n = 4 mice/group from three independent experiments. Scale bars, 100 μm.

Comment 3: Similarly, the authors show that *Vegfc* is induced by Lats DKO and then go on to show that interference with *Vegfc*:*Vegfr3* signaling using a AAV–mVEGFR34-7-Ig strategy to express the extracellular domain of *Vegfr3*, suppresses the lacteal phenotype. This is a very elegant experiment that certainly supports the model. It would be ideal if the authors also tested if knock out of *Vegfc* floxed allele in *Pdgfrb*-Cre expressing cells suppresses the Lats DKO phenotype.

Response: While we agree that it would be ideal to test whether deletion of *Vegfc* allele in *Pdgfrb*-cre expressing cells suppresses the *Lats1/2*-KO-induced phenotypes, we would like to mention that we could not perform this experiment due to the absence of *Vegfc*-floxed mouse in our laboratory.

Nevertheless, to support our claim of upregulated VEGF-C in PDGFR β ⁺ cells by *Lats1/2*-KO, we performed single-molecule fluorescence *in situ* hybridization (smFISH) in *tdTomato*^{rPBC} and *Lats1/2*^Δ-*tdTomato*^{rPBC} mice. Through smFISH analysis, we confirmed that *Vegfc* expression was upregulated in PDGFR β ⁺ cells of *Lats1/2*^Δ-*tdTomato*^{rPBC} mice compared with *tdTomato*^{rPBC} mice (Fig. 7c). Moreover, the expression of *Vegfc* was mainly upregulated in PDGFR β ⁺ cells over other cell types after *Lats1/2* depletion (Fig. 7c). These findings suggest that *Vegfc* upregulation in PDGFR β ⁺ cells by *Lats1/2* depletion induces aberrant lacteal phenotypes. We included these additional results and their descriptions into the revised manuscript accordingly as below.

Results (Page 18) In addition, through single-molecule *in situ* hybridization (smFISH) of *Vegfc*, we observed that the expression of *Vegfc* is upregulated in *Lats1/2*^Δ-*tdTomato*^{rPBC} mice compared with *tdTomato*^{rPBC} mice (Fig. 7c).

Figure 7. vFB1-3 are distinct cell types secreting VEGF-C. c, Representative images of *Vegfc* single-molecule fluorescence *in situ* hybridization (smFISH) in intestinal villi of *tdTomato*^{rPBC} and *Lats1/2*^Δ-*tdTomato*^{rPBC} mice. White dotted boxes are magnified in the right panels. Scale bars, 20 μ m.

Comment 4: The authors go on to nicely demonstrate regulation of this pathway by mechanotransduction mechanisms and conclude with single cell analysis of *Pdgfb* lineage-traced cells to try and identify the key fibroblast subtype that mediates *Vegfc* expression and lacteal morphogenesis. They identify novel subsets that may be the key targets, but this line of investigation ends abruptly, and the analysis is a bit superficial in my opinion. In its current form I am not sure it adds a huge amount to the manuscript as it opens up more questions and there are many more analyses required. A number of questions need addressing:

Comment 4-1: it would be useful to know how many stromal cells are *Pdgfrb*-positive. Comparison to a recently published single cell profile of intestinal mesenchyme would be useful here (PMID: 31953387).

Response: We applied the same single-cell analysis as was performed with our dataset on the dataset suggested by the reviewer for two purposes. First, considering Comment 4-1, we investigated the percentage of stromal cells expressing *Pdgfrb*. Second, in response to Comment 4-5, we proceeded to interrogate whether we could also identify the fibroblast clusters vFB1-5 in the public dataset as well. Therefore, we performed unsupervised clustering of the public dataset and visualized *Pdgfrb* as well as marker genes presented by both the previous study and our study (For Reviewer Only Fig. 3a,b).

First of all, we detected that the largest cluster had a fair distribution of *Pdgfrb* expression (For Reviewer Only Fig. 3a). Although the expression of *Pdgfrb* seems low, we think it is due to the sparse nature of the single cell RNA data and the limitations of detecting features during library generation. Secondly, in the previous study, the authors identified various populations including Stromal cells_ *C1qtnf3* high, Stromal cells_ *Dkk2* high, Stromal Cells_ *Cxcl5*, Stromal cells_ *Ackr4*_high and Stromal cells_ *Ednrb* high (Cluster 1, 2, 3, 4, and 5 in UMAP visualization, respectively) and characterized them by markers *C1qtnf3*, *Dkk2*, *Cxcl5*, *Ackr4*, and *Ednrb*, respectively. When we compared the expression pattern of the markers that we presented (*Serpine1*, *Dkk2*, *P2rx1*, *Ackr4*, and *Bmp7* for vFB1, vFB2, vFB3, vFB4, and vFB5, respectively), we could see that either the same gene was used to mark a specific population (*Dkk2* and *Ackr4*) or the markers had very similar expression patterns by UMAP visualization (*C1qtnf3* and *Serpine1*, *Cxcl5* and *P2rx1*, and *Ednrb* and *Bmp7*). Thus, our clustering analysis of the public dataset and comparison with our data revealed a high similarity in fibroblast identity. (For Reviewer Only Fig. 3b)

Collectively, this computational analysis along with our staining data provided in response to reviewer's comment 4-3 and 4-4 strongly support that vFB1-3 identified in this study are indeed independent subsets of fibroblast that contribute to the maintenance of lacteal structure.

For Reviewer Only, Fig. 3 Validation of identity of intestinal fibroblast subsets through analysis of publicly available single-cell dataset of small intestinal mesenchyme.

a, Visualization of UMAP of unsupervised clustering of publicly available intestinal mesenchyme single cell dataset (left) and expression of *Pdgfrb* across multiple fibroblast subsets (right). **b**, Visualization of expression of marker genes in the clustering of publicly available dataset. Marker genes for stromal subsets from the public data (above) and marker genes for five fibroblast subsets identified in our study (below). Note the marker genes from two independent dataset exhibit cluster-specific expression patterns, suggesting similar identities of fibroblast subsets.

Comment 4-2: the authors identify a vFB1 population that are very similar to vFB2, except that vFB1 are marked by *Fos* and *Jun*. vFB1 may reflect a phenotypic artefact of cell stress induced by dissociation. Analysis of stress genes should be included and the existence of this cell type clarified by testing if vFB1 exist *in vivo*.

Response: We appreciate this constructive comment. We have also noticed distinct expression of stress genes or immediate early genes (IEGs) in vFB1, and acknowledged that such observation could result from an artifact arising from cell dissociation procedures. However, as would be in Comment 4-3, through *in vivo* immunofluorescence staining, we were able to observe *Pdgfrb*⁺ *Fosb*⁺ populations in the small intestine LP, regardless of the cell dissociation processes (Fig. 7b).

Moreover, from our single-cell analysis we also observed various marker genes distinctively expressed in vFB1 that are not related to IEGS such as *Serpine1*, *C1qtnf3* and *B4galt3*. In the light of these evidences, we believe that vFB1 and vFB2 are indeed distinct subsets of fibroblasts residing in the intestine lamina propria.

Comment 4-3: the authors show some in situ studies but the quality is poor and the key question; are they distinct cell types? needs co-staining with multiple markers and imaging at single cell resolution.

Response: This is valid point to be addressed. Please note that this issue was also discussed in Reviewer 1's Comment 7 and Reviewer 4's General comments. We performed additional immunofluorescence co-staining of the distinctively expressed marker genes identified from scRNA-Seq.

We focused on distinguishing the newly identified vFB1-3 populations rather than previously described submucosal fibroblast (FB4), subepithelial fibroblast (FB5), smooth muscle cell, and mural cell populations. Although vFB1, 2, and 3 shared the expression of several genes such as *Serpina3N*, we were able to distinguish vFB3 from vFB1 and 2 by the specific expression of *P2X1* in vFB3 (Fig. 7a). In addition, while vFB1 and 2 commonly expressed *Shisa3*, vFB1 distinctly *Fosb* (Fig. 7b). Through additional immunofluorescence studies, we were able to validate existence and distinctive nature of fibroblast heterogeneity composing the small intestine lamina propria. We included these additional results and their descriptions into the revised manuscript accordingly as below.

Results (Page 18). To define the location of these newly identified subsets of fibroblasts, we examined and validated the expressions of markers that are specific to each cluster (Fig. 6h,i). Of note, vFB1-3 were distributed in close proximity to the lacteal, while *Ackr4*⁺ FB4 and *Sox6*⁺ FB5 were localized around the submucosal and subepithelial regions, respectively (Fig. 6h,i), as previously reported^{12, 41}. **vFB1-3 were distinguished by specific markers: *P2X1* was exclusively expressed by vFB3 but not by vFB1-2; *Fosb* was specifically expressed by vFB1 but not by vFB2 (Fig. 7a,b).**

Figure 7. vFB1-3 are distinct cell types secreting VEGF-C. **a**, Representative images of vFB3 (P2X1⁺Serpina3n⁺PDGFRβ⁺) and vFB1-2 (P2X1⁻Serpina3n⁺PDGFRβ⁺) in WT mouse. White and yellow dotted boxes are magnified in the upper right and lower right panels, respectively. Red and yellow arrowheads indicate vFB3 and vFB1-2, respectively. Scale bar, 20 μm. **b**, Representative images of vFB1 (Serpina3n⁺Fosb⁺Shisa3⁺PDGFRβ⁺) and vFB2 (Serpina3n⁺Fosb⁻Shisa3⁺PDGFRβ⁺) in WT mouse. White and yellow dotted boxes are magnified in the upper right and lower right panels, respectively, for each image. White and yellow arrowheads indicate vFB1 and vFB2, respectively. Scale bars, 20 μm.

Comment 4-4: the authors should validate *in vivo* by *in situ* or IHC that *Vegfc* is regulated by *Lats1/2* and *Yap/Taz* in the novel vFB populations to distinguish if all three vFBs regulate *Vegfc*

Response: Unfortunately, there are no validated VEGF-C antibody for IHC. Instead, we performed *Vegfc* single-molecule fluorescence *in situ* hybridization (smFISH) in vFB1-3 populations. First, we observed that expression of *Vegfc* is upregulated in PDGFRβ⁺ cells of *Lats1/2*^{iΔ}-*tdTomato*^{rPBC} mice compared with *tdTomato*^{rPBC} mice (Fig. 7c). Specifically, *Vegfc* expression is upregulated in Shisa3⁺ PDGFRβ⁺ vFB1-2 and P2X1⁺ PDGFRβ⁺ vFB3 of *Lats1/2*^{ΔPBC} mice compared with WT mice (Fig. 7d). Thus, *Vegfc* expression is upregulated in all three vFB1-3 populations. We included these additional findings and their descriptions into the revised manuscript as below.

Results (Page 18) In addition, through single-molecule *in situ* hybridization (smFISH) of *Vegfc*, we observed that the expression of *Vegfc* is upregulated in *Lats1/2*^{iΔ}-*tdTomato*^{rPBC} mice compared with *tdTomato*^{rPBC} mice (Fig. 7c). Moreover, we performed smFISH of *Vegfc* with immunofluorescence staining of vFB1-3-specific markers. *Vegfc* expression increased in vFB1-2 (Shisa3⁺PDGFRβ⁺ cells) and vFB3 (P2X1⁺PDGFRβ⁺ cells) in *Lats1/2*^{ΔPBC} mice compared with WT mice (Fig. 7d).

Figure 7. vFB1-3 are distinct cell types secreting VEGF-C.

c, Representative images of *Vegfc* single-molecule fluorescence *in situ* hybridization (smFISH) in intestinal villi of *tdTomato*^{rPβC} and *Lats1/2*^{iΔ}-*tdTomato*^{rPβC} mice. White dotted boxes are magnified in the right panel. Scale bars, 20 μm. **d**, Representative images of *Vegfc* smFISH in vFB1-3 of WT and *Lats1/2*^{iΔ}-*tdTomato*^{rPβC} mice. White and yellow dotted boxes are magnified in the upper right and lower right panels, respectively, for each image. Scale bars, 20 μm.

Comment 4-5: vFB1-3 don't seem very distinct in the UMAP. What do the distinct vFB1-3 signature gene profiles look like across all cells? My concern is whether these are really distinct cell types as proposed by the authors. For example, Vegfc looks like it marks most of vFB3, but only subsets of vFB1-2 populations. More detailed clustering of the vFB1-4 populations would be helpful in defining what's going on with these populations.

Response: This is an important point to be addressed. As shown in Fig. 6d,e, we observed several genes that are distinctively expressed by different vFB populations. To deliver more convincing evidences on the existence and distinctiveness of vFB populations, we performed additional immunofluorescence as discussed in Comment 4-3 and Fig. 7a,b. We were able to distinguish various vFB populations using the marker genes we have identified. Moreover, related to Comment 4-1, comparing our results with a publicly available dataset revealed highly correlative populations constituting the intestine lamina propria (For Reviewer Only Fig. 3). Thus, although the developmental processes such as differentiation/transitions of vFBs require further investigation, we suggest that these populations harbor not only discrete transcriptomic profiles but also distinct cellular localizations *in vivo*, indicating their distinctive nature in the intestinal milieu.

Comment 4-6: a subset of FB4 (Grem+;Ackr4+) are also marked by Vegfc, so FB4 might contribute to lacteal integrity, not just FB1-3 as proposed by the authors. This needs to be examined more carefully.

Response: As the reviewer mentioned, subsets of FB4 also showed *Vegfc* expression. However, the FB4 subset are mainly located in the submucosal layer (Fig. 6i) (Thomson, C.A. et al., *J Immunol*, 2018). In addition, even though FB4 showed some *Vegfc* expression, the proportion of *Vegfc*⁺ FB4 cells were relatively lower than that of vFB1-3. Therefore, considering vFB1-3's close proximity to the lacteal and the high proportion of *Vegfc*⁺ cells, we propose that vFB1-3 are the main regulators of lacteal integrity.

Comment 4-7: what's the cell type abundance in WT versus Lats DKO?

We calculated the cell-cluster abundance (percentage from total) for all 4 datasets we analyzed in this study. We fully acknowledge that the populations of cell clusters could be influenced by the methods used for library preparation. However, we observed no significant difference in the composition of PDGFR β ⁺ cell populations constituting the small intestine lamina propria in computational analyses.

Response

For Reviewer Only, Fig. 4 Compositional analysis of PDGFR β ⁺ cells constituting the small intestine lamina propria. Pie chart visualizing the abundance of PDGFR β ⁺ cell clusters in each of the four datasets—two replicates of WT and *Lats1/2*-KO.

***References for responses to comments of Referee #3**

Carolyn A Thomason et al., (2018). Expression of the atypical chemokine receptor ACKR4 identifies a novel population of intestinal submucosal fibroblasts that preferentially expresses endothelial cell regulators. *J Immunol*. Jul 1;201(1)215-229.

Choi SY et al., (2020). YAP/TAZ direct commitment and maturation of lymph node fibroblastic reticular cells. *Nat Commun*. Jan 24;11(1):519.

Kato K et al., (2018). Pulmonary pericytes regulate lung morphogenesis. *Nat Commun*. Jun 22;9(1):2448.

Kim JE et al., (2020). Single cell and genetic analyses reveal conserved populations and signaling mechanism of gastrointestinal stromal niches. *Nat Commun*. Jan 17;11(1):334.

Park DY et al., (2017). Plastic roles of pericytes in the blood-retinal barrier. *Nat Commun*. May 16;8:15296.

Referee #4

General Comments :

The authors have conducted a well-conceived and thorough study into the role of YAP/TAZ signaling in intestinal stromal cells. This is a very nice study with a substantial amount of evidence supporting the role of villi fibroblasts in relation to lacteal integrity.

Of note, the application of scRNA-seq to examine cell heterogeneity allowed the detection of several important subtypes with apparently different functional roles. This information was used to define the function and location of several different, and novel, cell types. This detail is a major advancement that will promote in-depth studies of these intestinal cells in both health and disease.

Overall, this was an enjoyable paper to read and I congratulate the authors on their efforts.

Despite the massive amount of work, I can't help but feel that the scRNA-seq was merely an afterthought and there seems to be little attempt to validate any of the novel subpopulations using orthogonal methods. My main constructive criticism here is that it is crucial to validate any expression data, particularly scRNA-seq, but not much was done to validate the cell types identified here. There are multiple robust wet-lab methods available to define the expression of population-specific marker genes at the single cell level - smRNA-FISH being a popular alternative to standard IHC.

Another suggestion is to consider silencing or deleting these subpopulations, either as a cell line or in vivo, to better understand their overall effect.

Response: Thank you for this constructive comment. Please note that this issue was also discussed in Reviewer 1's comment 7 and Reviewer 3's comment 4-3. We performed additional immunofluorescence co-staining of the distinctively expressed marker genes identified from scRNA-Seq.

We focused on distinguishing the newly identified vFB1-3 populations rather than previously described submucosal fibroblast (FB4), subepithelial fibroblast (FB5), smooth muscle cell, and mural cell populations. Although vFB1, 2, and 3 shared the expression of several genes such as *Serpina3N*, we were able to distinguish vFB3 from vFB1 and 2 by the specific expression of *P2X1* in vFB3 (Fig. 7a). In addition, while vFB1 and 2 commonly expressed *Shisa3*, vFB1 distinctly *Fosb* (Fig. 7b). Through additional immunofluorescence studies, we were able to validate existence and distinctive nature of fibroblast heterogeneity composing the small intestine lamina propria. We included these additional results and their descriptions into the revised manuscript accordingly as below.

Results (Page 18) To define the location of these newly identified subsets of fibroblasts, we examined and validated the expressions of markers that are specific to each cluster (Fig. 6h,i). Of note, vFB1-3 were distributed in close proximity to the lacteal, while $Ackr4^+$ FB4 and $Sox6^+$ FB5 were localized around the submucosal and subepithelial regions, respectively (Fig. 6h,i), as previously reported^{12, 41}. vFB1-3 were distinguished by specific markers: P2X1 was exclusively expressed by vFB3 but not by vFB1-2; *Fosb* was specifically expressed by vFB1 but not by vFB2 (Fig. 7a,b).

Figure 7. vFB1-3 are distinct cell types secreting VEGF-C. **a**, Representative images of vFB3 ($P2X1^+Serpina3n^+PDGFR\beta^+$) and vFB1-2 ($P2X1^-Serpina3n^+PDGFR\beta^+$) in WT mouse. White and yellow dotted boxes are magnified in the upper right and lower right panels, respectively. Red and yellow arrowheads indicate vFB3 and vFB1-2, respectively. Scale bar, 20 μ m. **b**, Representative images of vFB1 ($Serpina3n^+Fosb^+Shisa3^+PDGFR\beta^+$) and vFB2 ($Serpina3n^+Fosb^-Shisa3^+PDGFR\beta^+$) in WT mouse. White and yellow dotted boxes are magnified in the upper right and lower right panels, respectively, for each image. White and yellow arrowheads indicate vFB1 and vFB2, respectively. Scale bars, 20 μ m.

There are neither a Cre-inducible mouse that can specifically target the fibroblast subpopulations nor an *in vitro* system that can fully recapitulate the intestinal microenvironment consisting of lacteals and diverse fibroblasts in our laboratory. Therefore, we are planning to investigate this further in a separate study to identify the functions and features of the distinct subpopulations that we validated in Fig. 7 when the necessary genetically modified mice are available.

Detailed Comment 1: Authors need to please check the methods for the genomics components of the study. As far as this reviewer is aware, the Illumina Hi-Seq-X is not compatible with RNA-seq libraries of any kind - it is more likely that a different instrument was used, such as a HiSeq 4000 or NovaSeq. Following this, please add more details about the sequencing run configuration, sequencing read depth per library and the subsequent read depth per cell. This detail is important to understand the quality of the dataset being presented.

Response: We appreciate this thoughtful comment. While we acknowledge that Illumina does not officially support Hiseq-X for sequencing RNA libraries, there is a report (Zhao *et al.*, *Cell Stem Cell*, 2018) using Hiseq-X for single-cell library generation. In fact, we performed a pilot study using other Hiseq platforms including Hiseq-2500. However, the sequencing qualities were better when produced by Hiseq-X. The detailed configurations were 150 paired-end read, with each library ran in single lanes. The consequent read library depths were 100~110G (per library) and mean read depth per cell was 79,252 through CellRanger v3.

Detailed Comment 2: I noted the lower threshold of 'detected genes per cell' as 1000, which is actually quite high. How was this value arrived at? In typical situations, 200-500 is sufficient. I'm curious to know whether the extra 500 genes per cell would influence the cluster cell-type classifications. Have you tried to alter these thresholds and look at the outcomes here?

Response: The minimum threshold for detected genes per cell was derived by manual inspection of histograms for each library (For Reviewer Only, Fig. 5). We observed a small peak of cells with low detected genes, thus regarded them as low quality cells. We applied 1000 as the minimum threshold uniformly across datasets and removed a mean of 4.18% of total cells for each dataset. In light of reviewer's comments, we tried applying the minimum threshold as 500 genes per cell, which added only a small number of cells to each dataset (15, 9, 17, and 39 cells, respectively).

For Reviewer Only, Fig. 5 Distribution of the number of detected genes per cell across four datasets. Histogram depicting the distribution of the number of genes detected per cell for each four datasets. The vertical line in the histogram is located at the threshold point (1000 genes). The number in top left corner denotes the number of cells below and above the threshold 1000.

Moreover, unsupervised clustering showed that changing the threshold did not have a large influence on the cluster identifications (cells that were added by changing the threshold are in red) (For Reviewer Only, Fig. 6). In addition, we found that the number of genes detected did not drive clustering of single cells.

For Reviewer Only, Fig. 6 Visualization of UMAP of unsupervised clustering using threshold of detected genes per cell as 500. Unsupervised clustering after incorporation of 80 cells with detected genes per cell in the range of 500~1000. The added cells are marked in red (left). Visualization of the number of features detected per cell (right). Note the number of genes detected does not affect the clustering

Detailed Comment 3: Regarding the differential expression tests, how did the authors settle on using MAST as the DE test method? Were other methods also used? If so, how did the results appear?

Response: We utilized MAST to identify differentially expressed genes (DEGs) which accounts for potential dropouts in the gene expression data. Comparison of multiple DE testing methods (Soneson *et al.*, *Nat methods*, 2018) indicated that MAST shows the best performance in single cell DE testing. In light of the reviewer's comment, we also used other DE testing methods implemented in the Seurat package. We identified marker genes for each cluster using Wilcoxon Rank Sum test ("wilcox"), likelihood-ratio test ("bimod"), student's t-test ("t"), negative binomial generalized linear model test ("negbiom"), and DESeq2. After identification, the top 100 genes with highest average logFC were defined as marker genes. The overlap of identified marker genes was visualized by calculating jaccard index against markers identified by MAST. We observed that most of the identified DEGs were in common regardless of the DE testing method utilized.

For Reviewer Only, Fig. 7 Various differential expressed gene testing methods reveal similar gene sets that are specifically expressed across clusters. Heatmap visualizing the jaccard index of top 100 differentially expressed genes for PDGFR β ⁺ clusters identified by five different DE testing methods.

Detailed Comment 4: It would be fascinating to use a knockout mouse for any one of those key marker genes to see if there is an obvious phenotypic effect on the lacteal. The model(s) could then be used for additional scRNA-seq comparisons and/or methods such as single molecule RNA-FISH to more clearly show how these populations are functioning.

Response: We agree that this would be a great addition to this study. However, as we responded in General comments, generating a Cre-inducible mouse specific for the subpopulations during the revision period is not possible. We are planning to do this as a separate study.

Detailed Comment 5: One important avenue that was not explored in this work was that of the human translation. Are there any human single cell datasets or tissue samples available that any of these novel populations can also be identified in? I believe this to be probably one of the most important aspects of this study; should these populations be identified and functionally understood in humans, I expect there would be huge implications for GI health.

Response: We fully agree with this comment. Unfortunately, however, no human single-cell dataset illuminating small intestinal fibroblasts is publically available. In fact, it is hard to obtain a healthy portion of human small intestine since a resection of small intestine is rare, while that of large intestine is common. Nevertheless, in response to this comment, we acquired a single-cell dataset of human colon stromal cells (accession: GSE114374) and analyzed it accordingly. We were able to detect some heterogeneity in fibroblasts of colonic stroma, but it did not show high correlation with the sub-populations that we identified (data not shown). We believe that perhaps the distinct function and micro-environment between small intestine and colon are responsible for the differences in composition of stromal cells in the lamina propria.

Detailed Comment 6: Finally, please make the datasets available to the public - I note that you have not yet uploaded the data to a repository, or made it available to the reviewers.

Response: We uploaded the datasets to GEO in advance, and the datasets are scheduled to be released on Dec 31, 2020. Therefore, we provide a secure token yhcfeleshpcnhop in order to make it available to the reviewers.

***References for responses to comments of Referee #4**

Kinchen J et al., (2018). Structural Remodeling of the Human Colonic Mesenchyme in Inflammatory Bowel Disease. *Cell*. Oc4;175(2):372-386

Soneson C et al., (2018). Bias, robustness and scalability in single-cell differential expression analysis. *Nat Methods*. Feb 26;15(4):255-261

REVIEWERS' COMMENTS:

Reviewer #4 (Remarks to the Author):

Thank you for addressing my concerns. I am satisfied with the extra data that you have supplied and have no further comments.

Reviewer #5 (Remarks to the Author):

Review for Nature Communications of manuscript NCOMMS-19-40090A entitled:

Distinct fibroblast subsets regulate lacteal integrity through YAP/TAZ-induced VEGF-C in intestinal villi

Seon Pyo Hong, Myung Jin Yang Hyunsoo Cho, Intae Park, Hosung Bae, Kibaek Choe, Sang Heon Suh, Ralf H. Adams, Kari Alitalo, Daesik Lim and Gou Young Koh

1. The authors effectively address the reviewer's comments in their detailed point-by-point responses. The responses are largely clear, appropriate, and responsive, and the new data presented in these responses are relevant. Where changes are made in the text and new data are added to figures, the responses make the manuscript stronger and more compelling and adequately resolve the issues raised.

2. However, in multiple cases, the point-by-point responses address the reviewer's comment, without making corresponding changes in the manuscript text or figures. The responses themselves are acceptable, but they do not resolve the issue(s) in the manuscript that led to the reviewer's comment in the first place. Therefore, there is nothing in the manuscript to prevent other readers from raising the same issues. The concern applies to the author's written responses and to new data presented as "For Reviewer Only".

3. This concern applies to Reviewer #1 Comments 2-5. The straightforward remedy is to address the reviewer's comments by making corresponding changes in the Results and/or Discussion and by adding the new data to main or extended figures. References cited in point-by-point responses should also be added.

4. The following changes in the manuscript text, figures, and references are recommended to incorporate the authors' point-by-point responses to Reviewer #1 comments 2-5:

- a. Comment 2: add text and figure
- b. Comment 3: add text and reference (Suh et al. 2019)
- c. Comment 4: add text
- d. Comment 5: add text, figure, and reference (Tropini et al. 2018)

5. The same concern also applies to responses to Reviewer #3 Comments 1 and 4-1, 4-2, 4-5, 4-6, 4-7.

6. Similarly, the concern applies to the responses to Reviewer #4 Detailed Comments 1-6.

Reviewer #6 (Remarks to the Author):

NOTE: this reviewer has been recruited to judge on the referee report of a previous reviewer who was not anymore available. Therefore, for the sake of the revieweing process and of its fairness, this reviewer will only judge on whether the authors answered the concerns previously raised.

Comment 1: the literature in support is exhaustive and indicates the specificity of the driver.

Comment 2: the data on the *lats1/2*+YAP/TAZ knockouts, provided during revision, is well above the normal standards of revision. This adds a very significant piece of information to the manuscript.

Comment 3: I think the previous reviewer asked a bit too much. He/she asked for two more genetic combinations during revision (comment 2 and 3), which can be unfair due to rederivation/crossing times. The data with AAV is already sufficient in my opinion.

Comment 4: The authors made effort to clarify the issues on single-cell sequencing along the lines indicated by the reviewer, and in my opinion this data is now suitable for publication. I would however strongly recommend to include all data "for reviewers only" in the supplementary material, as this will reinforce the analysis and the future comparisons other scientists may want to perform.

Minor additional note 1: in the paragraph on mechanotransduction, it would be worth introducing it by clarifying this represents an independent experimental means, based on a potentially relevant physiological stimulus (i.e. stretching), to study the role of YAP/TAZ activation in these cells. This cannot be claimed however to represent a full demonstration that the phenotypes seen by *LATS1/2* inactivation can also occur in the real intestine BECAUSE OF mechanotransduction, since data are missing to support this interesting idea. Moreover, in principle, physiological mechanical stimulation of YAP/TAZ will not induce these phenotypes, because these phenotypes are not observed in the normal intestine. On the same line, using osmotic pressure to change cell mechanics may be useful experimentally, but is even harder to accept that a normal intestine is naturally subjected to such strong osmotic stresses. This should be clarified.

Minor additional note 2: ISC is already established in the literature for Intestinal Stem Cells. Given that the paper is on the intestine, but not on stem cells, I would recommend to use an alternative name for stromal cells, and to use it in a coherent manner throughout the paper (in principle, these are the same cells in which *LATS1/2* are ablated, so pericytes).

Point-by-Point Response to Reviewers Comments

Comments of Reviewer #5:

Comment 1: The authors effectively address the reviewer's comments in their detailed point-by-point responses. The responses are largely clear, appropriate, and responsive, and the new data presented in these responses are relevant. Where changes are made in the text and new data are added to figures, the responses make the manuscript stronger and more compelling and adequately resolve the issues raised.

Comment 2: However, in multiple cases, the point-by-point responses address the reviewer's comment, without making corresponding changes in the manuscript text or figures. The responses themselves are acceptable, but they do not resolve the issue(s) in the manuscript that led to the reviewer's comment in the first place. Therefore, there is nothing in the manuscript to prevent other readers from raising the same issues. The concern applies to the author's written responses and to new data presented as "For Reviewer Only".

Comment 3: This concern applies to Reviewer #1 Comments 2-5. The straightforward remedy is to address the reviewer's comments by making corresponding changes in the Results and/or Discussion and by adding the new data to main or extended figures. References cited in point-by-point responses should also be added.

Comment 4: The following changes in the manuscript text, figures, and references are recommended to incorporate the authors' point-by-point responses to Reviewer #1 comments 2-5:

a. **Comment 2:** add text and figure

b. **Comment 3:** add text and reference (Suh et al. 2019)

c. **Comment 4:** add text

d. **Comment 5:** add text, figure, and reference (Tropini et al. 2018)

5. The same concern also applies to responses to Reviewer #3 Comments 1 and 4-1, 4-2, 4-5, 4-6, 4-7.

6. Similarly, the concern applies to the responses to Reviewer #4 Detailed Comments 1-6.

Response to comments 1-6: We added "For Reviewer Only" figures and made changes in the text accordingly. We have incorporated For Reviewer Only, Fig. 1, illustrating no significant changes in numbers of F4/80⁺ macrophages as Supplementary Fig.4. Also, For Reviewer Only, Fig. 2 demonstrating translocation of

YAP/TAZ upon osmotic stress was added to our manuscript as Supplementary Fig. 13. Corresponding explanation of the data and references were added to our manuscript. The compositional analysis data shown in For Reviewer Only Fig. 4 were incorporated into Supplementary Fig. 13. For Reviewer Only figures regarding the determination of threshold of 'detected genes per cell' (For Reviewer Only, Fig.5-6) were also added to manuscript as Supplementary Fig. 14. Of note, For Reviewer Only Fig. 6 was substituted to more relevant data visualizing the number of genes detected per cell projected on 2D UMAP plots of WT and KO datasets. Additionally, we wish have not included two For Reviewers Only Figures as we believe that they are not suitable to our manuscript. First, the comparative analysis using publicly available data as shown in For Reviewer Only, Fig. 3, we believe that our data alone are sufficient to validate the clustering and cell type identifications, and thereby wish to exclude figures generated using public datasets. Lastly, we wish to exclude For Reviewer Only, Fig 7 as comparison of various differential testing methods have not showed differences in identification of DEGs.

Results (Page 9) In addition, F4/80+ macrophages, previously suggested as source of VEGF-C³⁵, showed no significant changes in their numbers between WT and *Lats1/2*^{ΔPβC} mice. (Supplementary Fig. 4a-c).

Supplementary Fig. 4 Flow cytometric analysis of MHCII⁺ F4/80⁺ macrophages in WT and *Lats1/2*^{ΔPβC} mice. **a**, Representative flow cytometric analysis with gating strategy

from the adult small intestine in WT and *Lats1/2*^{ΔPBC} mice. **b,c**, Representative flow cytometric analysis and comparison of MHCII⁺ F4/80⁺ macrophage gated on DAPI⁻ CD45⁺ cells from whole small intestine of WT and *Lats1/2*^{ΔPBC} mice. Each dot indicates value from *n* = 4 mice/group. Horizontal bars indicate mean ± SD and *P* value versus WT by two-tailed Mann-Whitney *U* test. n.s., not significant.

Results (Page 14) Furthermore, osmotic perturbation into the *ex vivo* culture of mouse small intestine and *in vivo*⁴² promoted cytoplasmic translocations of YAP/TAZ in PDGFRβ⁺ IntSCs (Supplementary Fig. 13a-d).

Supplementary Fig. 13 Osmotic stress regulates translocation of YAP/TAZ in PDGFRβ⁺ IntSCs *ex vivo* and *in vivo*. **c**, Diagram for inducing osmotic stress in small intestine of WT mice by administration of 20% PEG starting at 8 weeks for 7 days and their analyses at 9 weeks. **d**, Representative images and comparisons of YAP and TAZ expression and their localizations in PDGFRβ⁺ IntSCs of control or osmotic stress applied small intestine *in vivo*. Insets show magnified view. Similar findings were observed in four independent experiments using *n* = 4 mice. Scale bars, 25 μm.

Results (Page 17) When we measured the correlation of marker gene overlaps between them, all of the seven distinct PDGFR β^+ intSC clusters demonstrated highly correlated transcriptional profiles upon YAP/TAZ hyperactivation, **while no significant changes in cell type abundances were observed** (Supplementary Fig. 16c,d).

Supplementary Fig. 16 Unsupervised clustering on the integrated dataset of PDGFR β^+ intSCs from tdTomato^{rPBC} and Lats1/2 Δ -tdTomato^{rPBC} mice suggests that vFB1-3 secrete VEGF-C. d, Pie chart visualizing the abundance of PDGFR β^+ cell clusters in each of the four datasets—two replicates of WT and *Lats1/2*-KO.

Methods (Page 38) Cells with fewer than 1,000 detected genes or higher than 6,000 detected genes (considered as potential doublets) were excluded (Supplementary Fig. 17a).

Methods (Page 39) Clustering resolution parameters were set at the point where clusters showed highly distinct transcriptional profiles. **Clusterings were independent to the number of genes detected per cell** (Supplementary Fig. 17b).

Supplementary Fig. 17 Cell-level quality control of single-cell RNA sequencing datasets

a, Histogram depicting the distribution of the number of genes detected per cell for each four datasets. The cyan vertical line in the histogram is located at the threshold point (1000 genes). The number in top left corner denotes the number of cells below and above the threshold 1000. **b**, Visualization of number of genes detected per cell in *tdTomato*^{r β C} and *Lats1/2*^{i Δ} -*tdTomato*^{r β C} datasets, projected on the UMAP plot.

Comments of Reviewer #6:

Comment 1: the literature in support is exhaustive and indicates the specificity of the driver.

Comment 2: the data on the *lats1/2*+YAP/TAZ knockouts, provided during revision, is well above the normal standards of revision. This adds a very significant piece of information to the manuscript.

Comment 3: I think the previous reviewer asked a bit too much. He/she asked for two more genetic combinations during revision (comment 2 and 3), which can be unfair due to rederivation/crossing times. The data with AAV is already sufficient in my opinion.

Response to comments 1~3: We appreciate these supporting comments.

Comment 4: The authors made efforts to clarify the issues on single-cell sequencing along the lines indicated by the reviewer, and in my opinion this data is now suitable for publication. I would however strongly recommend to include all data “for reviewers only” in the supplementary material, as this will reinforce the analysis and the future comparisons other scientists may want to perform.

Response: Comparison of publicly available intestinal mesenchyme dataset to our datasets was solely aimed to validate our clustering and cell type identifications. Indeed, we identified cell populations with similar transcriptomic profiles in the two datasets. Moreover, through additional FISH experiments, we provided solid evidences on the existence and distinctiveness of cell types that we have identified. Therefore, we believe our data alone are sufficient to support our claim, and wish to exclude the new figures generated using publically available datasets from the manuscript and only include our data of compositional analysis in the supplementary material.

Results (Page 17) When we measured the correlation of marker gene overlaps between them, all of the seven distinct PDGFR β^+ intSC clusters demonstrated highly correlated transcriptional profiles upon YAP/TAZ hyperactivation, **while no significant changes in cell type abundances were observed** (Extended Data Fig. 13c,d).

Supplementary Fig. 13 Unsupervised clustering on the integrated dataset of PDGFR β^+ intSCs from *tdTomato*^{rPBC} and *Lats1/2* Δ -*tdTomato*^{rPBC} mice suggests that vFB1-3 secrete VEGF-C. d, Pie chart visualizing the abundance of PDGFR β^+ cell clusters in each of the four datasets—two replicates of WT and *Lats1/2*-KO.

Minor additional note 1: in the paragraph on mechanotransduction, it would be worth introducing it by clarifying this represents an independent experimental

means, based on a potentially relevant physiological stimulus (i.e. stretching), to study the role of YAP/TAZ activation in these cells. This cannot be claimed however to represent a full demonstration that the phenotypes seen by LATS1/2 inactivation can also occur in the real intestine BECAUSE OF mechanotransduction, since data are missing to support this interesting idea. Moreover, in principle, physiological mechanical stimulation of YAP/TAZ will not induce these phenotypes, because these phenotypes are not observed in the normal intestine. On the same line, using osmotic pressure to change cell mechanics may be useful experimentally, but is even harder to accept that a normal intestine is naturally subjected to such strong osmotic stresses. This should be clarified.

Response: We modified the text in response to your comment.

Discussion (Page 22) However, the resulting phenotypes of our knockout mice and the *in vitro* experiment performed to induce mechanical or osmotic stress are harsher than those observed in physiological conditions. ~~However, coupling our phenotypic observations to pathophysiological conditions needs further investigation since we used mouse models in which YAP/TAZ are artificially activated.~~

Minor additional note 2: ISC is already established in the literature for Intestinal Stem Cells. Given that the paper is on the intestine, but not on stem cells, I would recommend to use an alternative name for stromal cells, and to use it in a coherent manner throughout the paper (in principle, these are the same cells in which LATS1/2 are ablated, so pericytes).

Response: We changed ISC to intSC for intestinal stromal cells to avoid confusion with Intestinal Stem Cells.